# Mirror Descent Actor Critic via Bounded Advantage Learning

## Abstract

Regularization is a core component of recent Reinforcement Learning (RL) al-
gorithms. Mirror Descent Value Iteration (MDVI) uses both Kullback-Leibler
divergence and entropy as regularizers in its value and policy updates. Despite
its empirical success in discrete action domains and strong theoretical guarantees,
the performance improvement of a MDVI-based method over the entropy-only-
regularized RL is limited in continuous action domains. In this study, we propose
Mirror Descent Actor Critic (MDAC) as an actor-critic style instantiation of MDVI
for continuous action domains, and show that its empirical performance is signifi-
cantly boosted by bounding the values of actor's log-density terms in the critic's
loss function. Further, we relate MDAC to Advantage Learning by recalling that
the actor's log-probability is equal to the regularized advantage function in tabular
cases, and theoretically show that the error of optimal policy misspecification is
decreased by bounding the advantage terms.

## 1 Introduction

Model-free reinforcement learning (RL) is a promising approach to acquire reasonable controllers
in unknown environments. In particular, actor-critic methods are appealing because they can be
naturally applied to continuous control domains. Actor-critic algorithms have been applied in a range
of challenging domains including robot control (Smith et al., 2023), magnetic control of tokamak
plasmas (Degrave et al., 2022), and alignment of large language models (Stiennon et al., 2020).

Regularization is a core component of, not only such actor-critic methods, but also value-based
reinforcement learning algorithms (Peters et al., 2010; Azar et al., 2012; Schulman et al., 2015; 2017;
Haarnoja et al., 2017; 2018a; Abdolmaleki et al., 2018). Kullback-Leibler (KL) divergence and
entropy are two major regularizers that have been adopted to derive many successful algorithms.
In particular, Mirror Descent Value Iteration (MDVI) uses both KL divergence and entropy as
regularizers in its value and policy updates (Geist et al., 2019; Vieillard et al., 2020a) and enjoys
strong theoretical guarantees (Vieillard et al., 2020a; Kozuno et al., 2022). However, despite its
empirical success in discrete action domains (Vieillard et al., 2020b), the performance improvement
of a MDVI-based algorithm over an entropy-only-regularized RL is limited in continuous action
domains (Vieillard et al., 2022).

In this study, we propose Mirror Descent Actor Critic (MDAC) as a model-free actor-critic instantia-
tion of MDVI for continuous action domains, and show that its empirical performance is significantly
boosted by bounding the values of actor's log-density terms in the critic's loss function. To understand
the impact of bounding beyond just as an "implementation detail", we relate MDAC to Advantage
Learning (Baird, 1999; Bellemare et al., 2016) by recalling that the policy's log-probability is equal
to the regularized advantage function in tabular case, and theoretically show that the error of optimal
policy misspecification is decreased by bounding the advantage terms. Our analysis indicates that it is
beneficial to bound the log-policy term of not only the current state-action pair but also the successor
pair in the TD target signal.

**Related Works.** The key component of our actor-critic algorithm is to bound the log-policy terms
in the critic loss, which can be also understood as bounding the regularized advantages. Munchausen
RL clips the log-policy term for the current state-action pair, which serves as an augumented reward,
as an implementation issue (Vieillard et al., 2020b). Our analysis further supports the empirical

suceess of Munchausen algorithms. Zhang et al. (2022) extended AL by introducing a clipping strategy, which increases the action gap only when the action values of suboptimal actions exceed a certain threshold. Our bounding strategy is different from theirs in the way that the action gap is increased for all state-action pairs but with bounded amounts. Vieillard et al. (2022) proposed a sound parameterization of Q-function that uses log-policy. By consruction, the regularized greedy step of MDVI can be performed exactly even in actor-critic settings with their parameterization. Our study is orthogonal to theirs since our approach modifies not the parameterization of the critic but its loss function.

MDVI and its variants are instances of mirror descent (MD) based RL. There are substantial research efforts in this direction (Wang et al., 2019; Vaswani et al., 2022; Kuba et al., 2022; Yang et al., 2022; Tomar et al., 2022; Lan, 2023; Alfano et al., 2023). The MD perspective enables to understand the existing successful algorithms in a unified view, analyze such methods with strong theoretical tools, and propose a novel and superior one. This paper focuses on a specific choice of mirror, i.e. adopting KL divegence and entropy as regularizers, and provides a deeper understanding in this specific scope via a notion of *gap-increasing* Bellman operators.

It is well known that the log-policy terms in actor-critic algorithms often cause instability, since the magnitude of log-policy terms grow large naturally in MDP, where a deterministic policy is optimal. Recent RL implementations handle this problem by bounding the range of the standard deviation for Gaussian policies (Achiam, 2018; Huang et al., 2022). Beyond such an implementation detail, Silver et al. (2014) proposed to use deterministic policy gradient, which is a foundation of the recent actor-critic algorithms such as TD3 (Fujimoto et al., 2018). On the other hand, Iwaki & Asada (2019) proposed an implicit iteration method to stably estimate the natural policy gradient (Kakade, 2001), which also can be viewd as a MD-based RL method (Thomas et al., 2013).

**Contibutions.** Our contributions are summarized as follows: (1) we proposed MDAC, a model-free actor-critic instantiation of MDVI for continuous action domains, and showed that bounding the log-density terms in the critic's loss function significantly improves the performance of MDAC, (2) we theoretically analyzed the validity and the effectivness of the bounding strategy by relating MDAC to AL with bouded advantage terms, (3) we empirically explored what types of bounding functions are effective, and (4) we demonstrated that MDAC performs better than baseline algorithms in simulated benchmarks.

## 2 PRELIMINARY

**MDP and Approximate Value Iteration.** A Markov Decision Process (MDP) is specified by a tuple $(\mathcal{S}, \mathcal{A}, P, R, \gamma)$, where $\mathcal{S}$ is a state space, $\mathcal{A}$ is an action space, $P$ is a Markovian transition kernel, $R$ is a reward function bounded by $R_{\max}$, and $\gamma \in (0, 1)$ is a discount factor. For $\tau \geq 0$, we write $V_{\max}^{\tau} = \frac{R_{\max} + \tau \log |\mathcal{A}|}{1 - \gamma}$ (assuming $\mathcal{A}$ is finite) and $V_{\max} = V_{\max}^0$. We write $\mathbf{1} \in \mathbb{R}^{\mathcal{S} \times \mathcal{A}}$ the vector whose components are all equal to one. A policy $\pi$ is a distribution over actions given a state. Let $\Pi$ denote a set of Markovian policies. The state-action value function associated with a policy $\pi$ is defined as $Q^{\pi}(s, a) = \mathbb{E}_{\pi}\left[\sum_{t=0}^{\infty} \gamma^t R(S_t, A_t) | S_0 = s, A_0 = a\right]$, where $\mathbb{E}_{\pi}$ is the expectation over trajectories generated under $\pi$. An optimal policy satisfies $\pi^* \in \operatorname{argmax}_{\pi \in \Pi} Q^{\pi}$ with the understanding that operators are point-wise, and $Q^* = Q^{\pi^*}$. For $f_1, f_2 \in \mathbb{R}^{\mathcal{S} \times \mathcal{A}}$, we define a component-wise dot product $\langle f_1, f_2 \rangle = \left(\sum_a f_1(s, a) f_2(s, a)\right)_s \in \mathbb{R}^{\mathcal{S}}$. Let $P_{\pi}$ denote the stochastic kernel induced by $\pi$. For $Q \in \mathbb{R}^{\mathcal{S} \times \mathcal{A}}$, let us define $P_{\pi} Q = \left(\sum_{s'} P(s'|s, a) \sum_{a'} \pi(a'|s') Q(s', a')\right)_{s,a} \in \mathbb{R}^{\mathcal{S} \times \mathcal{A}}$. Furthermore, for $V \in \mathbb{R}^{\mathcal{S}}$ let us define $PV = \left(\sum_{s'} P(s'|s, a) V(s')\right)_{s,a} \in \mathbb{R}^{\mathcal{S} \times \mathcal{A}}$ and $P^{\pi} V = \left(\sum_a \pi(a|s) \sum_{s'} P(s'|s, a) V(s')\right)_s \in \mathbb{R}^{\mathcal{S}}$. It holds that $P_{\pi} Q = P\langle \pi, Q \rangle$. The Bellman operator is defined as $\mathcal{T}_{\pi} Q = R + \gamma P_{\pi} Q$, whose unique fixed point is $Q^{\pi}$. The set of greedy policies w.r.t. $Q \in \mathbb{R}^{\mathcal{S} \times \mathcal{A}}$ is written as $\mathcal{G}(Q) = \operatorname{argmax}_{\pi \in \Pi}\langle Q, \pi \rangle$. Approximate Value Iteration (AVI) (Bellman & Dreyfus, 1959) is a classical approach to estimate an optimal policy. Let $Q_0 \in \mathbb{R}^{\mathcal{S} \times \mathcal{A}}$ be initialized as $\|Q_0\|_{\infty} \leq V_{\max}$ and $\epsilon_k \in \mathbb{R}^{\mathcal{S} \times \mathcal{A}}$ represent approximation/estimation errors. Then, AVI can be written as the following abstract form:

$$\begin{cases} \pi_{k+1} \in \mathcal{G}(Q_k) \\ Q_{k+1} = \mathcal{T}_{\pi_{k+1}} Q_k + \epsilon_{k+1} \end{cases}.$$

**Regularized MDP and MDVI.** In this study, we consider the Mirror Descent Value Iteration (MDVI) scheme (Geist et al., 2019; Vieillard et al., 2020a). Let us define the entropy $\mathcal{H}(\pi) = -\langle \pi, \log \pi \rangle \in \mathbb{R}^{\mathcal{S}}$ and the KL divergence $D_{\mathrm{KL}}(\pi_1 \| \pi_2) = \langle \pi_1, \log \pi_1 - \log \pi_2 \rangle \in \mathbb{R}^{\mathcal{S}}_{\geq 0}$. For $Q \in \mathbb{R}^{\mathcal{S} \times \mathcal{A}}$ and a reference policy $\mu \in \Pi$, we define the regularized greedy policy as $\mathcal{G}^{\lambda,\tau}_{\mu}(Q) = \arg\max_{\pi \in \Pi} \left( \langle \pi, Q \rangle + \tau \mathcal{H}(\pi) - \lambda D_{\mathrm{KL}}(\pi \| \mu) \right)$. We write $\mathcal{G}^{0,\tau}$ for $\lambda = 0$ and $\mathcal{G}^{0,0}(Q) = \mathcal{G}(Q)$. We define the soft state value function $V(s) \in \mathbb{R}^{\mathcal{S}}$ as $V(s) = \langle \pi, Q \rangle + \tau \mathcal{H}(\pi) - \lambda D_{\mathrm{KL}}(\pi \| \mu)$, where $\pi = \mathcal{G}^{\lambda,\tau}_{\mu}(Q)$. Furthermore, we define the regularized Bellman operator as $\mathcal{T}^{\lambda,\tau}_{\pi|\mu} Q = R + \gamma P \left( \langle \pi, Q \rangle + \tau \mathcal{H}(\pi) - \lambda D_{\mathrm{KL}}(\pi \| \mu) \right)$. Given these notations, MDVI scheme is defined as

$$\begin{cases} \pi_{k+1} = \mathcal{G}^{\lambda,\tau}_{\pi_k}(Q_k) \\ Q_{k+1} = \mathcal{T}^{\lambda,\tau}_{\pi_{k+1}|\pi_k} Q_k + \epsilon_{k+1} \end{cases}, \tag{1}$$

where $\pi_0$ is initialized as the uniform policy.

Vieillard et al. (2020b) proposed a reparameterization $\Psi_k = Q_k + \beta\alpha \log \pi_k$. Then, defining $\alpha = \tau + \lambda$ and $\beta = \lambda/(\tau + \lambda)$, the recursion (1) can be rewritten as

$$\begin{cases} \pi_{k+1} = \mathcal{G}^{0,\alpha}(\Psi_k) \\ \Psi_{k+1} = R + \beta\alpha \log \pi_{k+1} + \gamma P \langle \pi_{k+1}, \Psi_k - \alpha \log \pi_{k+1} \rangle + \epsilon_{k+1} \end{cases}. \tag{2}$$

We refer (2) as Munchausen Value Iteration (M-VI). In the recursion (2), KL regularization is implicitly applied through $\Psi_k$ and there is no need to store $\pi_k$ for explicit computation of the KL term. Notice that the regularized greedy policy $\pi_{k+1} = \mathcal{G}^{0,\alpha}(\Psi_k)$ can be obtained analytically in discrete action spaces as $\left( \mathcal{G}^{0,\alpha}(\Psi_k) \right)(s,a) = \frac{\exp \Psi_k(s,a)/\alpha}{\langle \mathbf{1}, \exp \Psi_k(s,a)/\alpha \rangle} =: \left( \mathrm{sm}_\alpha(\Psi_k) \right)(s,a)$.

## 3 MIRROR DESCENT ACTOR CRITIC WITH BOUNDED BONUS TERMS

In this section, we introduce a model-free actor-critic instantiation of MDVI for continuous action domains, and show that a naive implementation results in poor performance. Then, we demonstrate that its performance is improved significantly by a simple modification to its loss function.

Now we derive Mirror Descent Actor Critic (MDAC). Let $\pi_\theta$ be a tractable stochastic policy such as a Gaussian with a parameter $\theta$. Let $Q_\psi$ be a value function with a parameter $\psi$. The functions $\pi_\theta$ and $Q_\psi$ approximate $\pi_k$ and $\Psi_k$ in the recursion (2), respectively. Further, let $\bar{\psi}$ be a target parameter that is updated slowly, that is, $\bar{\psi} \leftarrow (1-\kappa)\bar{\psi} + \kappa\psi$ with $\kappa \in (0,1)$. Now, we derive the losses for the actor $\pi_\theta$ and the critic $Q_\psi$. Let $\mathcal{D}$ be a replay buffer that stores past experiences $\{(s,a,r,s')\}$. We can derive online and off-policy losses from the recursion (2) by (i) letting the parameterized policy $\pi_\theta$ be represent the information projection of $\pi_k$ in terms of the KL divergence, and (ii) approximating the expectations using the transition samples drawn from $\mathcal{D}$:

$$L^Q(\psi) = \mathbb{E}_{\substack{(s,a,r,s') \sim \mathcal{D}, \\ a' \sim \pi_\theta(\cdot|s')}} \Big[ \big( \underbrace{r + \beta\alpha \log \pi_\theta(a|s) + \gamma \big( Q_{\bar{\psi}}(s',a') - \alpha \log \pi_\theta(a'|s') \big)}_{y(s,a,r,s',a')} - Q_\psi(s,a) \big)^2 \Big], \tag{3}$$

$$L^\pi(\theta) = \mathbb{E}_{s \sim \mathcal{D}} \Big[ D_{\mathrm{KL}} \big( \pi_\theta(a|s) \, \| \, \mathrm{sm}_\alpha(Q_\psi)(s,a) \big) \Big] = \mathbb{E}_{\substack{s \sim \mathcal{D}, \\ a \sim \pi_\theta(\cdot|s)}} \Big[ \alpha \log \pi_\theta(a|s) - Q_\psi(s,a) \Big]. \tag{4}$$

Though $\pi_\theta$ can be any tractable distribution, we choose commonly used Gaussian policy in this paper. We lower-bound its standard deviation by a common hyperparameter $\log \sigma_{\min}$, which is typically fixed to $\log \sigma_{\min} = -20$ (Huang et al., 2022) or $\log \sigma_{\min} = -5$ (Achiam, 2018). Although there are two hyperparameters $\alpha$ and $\beta$ originated from KL and entropy regularization, these hyperparameters need not to be tuned manually. We fixed $\beta = 1 - (1-\gamma)^2$ as the theory of MDVI suggests (Kozuno et al., 2022). For $\alpha$, we perform an optimization process similar to SAC (Haarnoja et al., 2018b). Noticing that the strength of the entropy regularization is governed by $\tau = (1-\beta)\alpha$, we optimize the following loss in terms of $\alpha$ by stochastic gradient descent (SGD) with $\bar{\mathcal{H}} = -\dim(\mathcal{A})$:

$$L(\alpha) = \mathbb{E}_{\substack{s \sim \mathcal{D}, \\ a \sim \pi_\theta(\cdot|s)}} \big[ -(1-\beta)\alpha \log \pi_\theta(a|s) - (1-\beta)\alpha\bar{\mathcal{H}} \big] = (1-\beta)\alpha \, \mathbb{E}_{s \sim \mathcal{D}} \big[ \mathcal{H}(\pi_\theta(\cdot|s)) - \bar{\mathcal{H}} \big]. \tag{5}$$

The reader may notice that (3) and (4) are nothing more than SAC losses (Haarnoja et al., 2018a;b) with the Munchausen augumented reward (Vieillard et al., 2020b), and expect that optimizing these losses results in good performance. However, a naive implementation of these losses leads to poor performance. The gray learning curve in Figure 1 is an aggregated learning result for 6 Mujoco environments with $\log \sigma_{\min} = -5$ [1]. The left column of Figure 2 compares the individual quantities in the TD target in loss (3) for the initial learning phase in `Walker2d-v4` and `HalfCheetah-v4`. To be precise, the means of the quantities in the sampled minibatchs are plotted. Clearly, the magnitude of the log-density terms get much larger

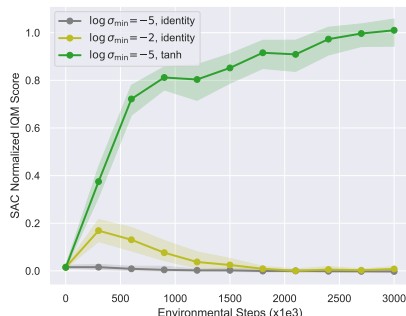

Figure 1: Effect of bounding $\log \pi_\theta$ terms.

than the reward quickly. We hypothesized that the poor performance of the naive implementation is due to this scale difference; the information of the reward is erased by the bonus terms. This explosion is more severe in the Munchausen bonus $\beta\alpha \log \pi_\theta(a|s)$ than the entropy bonus $\alpha \log \pi_\theta(a'|s')$, because while $a'$ is an *on-policy* sample from the current actor $\pi_\theta$, $a$ is an old *off-policy* sample from the replay buffer $\mathcal{D}$. Careful readers may wonder if the larger $\log \sigma_{\min}$ resolves this issue. The yellow learning curve in Figure 1 is the learning result for $\log \sigma_{\min} = -2$, which still fails to learn. The middle column of Figure 2 shows that the bonus terms are still divergent, and it is caused by the exploding behavior of $\alpha$. A naive update of $\alpha$ using the loss (5) and SGD is expressed as

$$\alpha \leftarrow \alpha + \frac{\rho(1-\beta)}{N} \sum_{n=1}^{N} \left(\log \pi_\theta(a_n|s_n) - \dim(\mathcal{A})\right),$$

where $\rho > 0$ is a step-size, $N$ is a mini-batch size and $a_n \sim \pi_\theta(\cdot|s_n)$. This expression indicates that, if the average of $\log \pi_\theta(a|s)$ over sampled mini-batches are bigger than $\dim(\mathcal{A})$, $\alpha$ keeps growing. Figure 2 indicates this phenomenon is indeed happening. We argue that, an unstable behavior of a single component ruins the other learning components through the actor-critic structure. The $\alpha \log \pi_\theta$ terms make $Q_\psi$ oscillatory, which hinders the optimization of the policy $\pi_\theta$ and the coefficient $\alpha$ through the losses (4) and (5). Then, $\alpha \log \pi_\theta$ terms explode gradually and ruins $Q_\psi$ again.

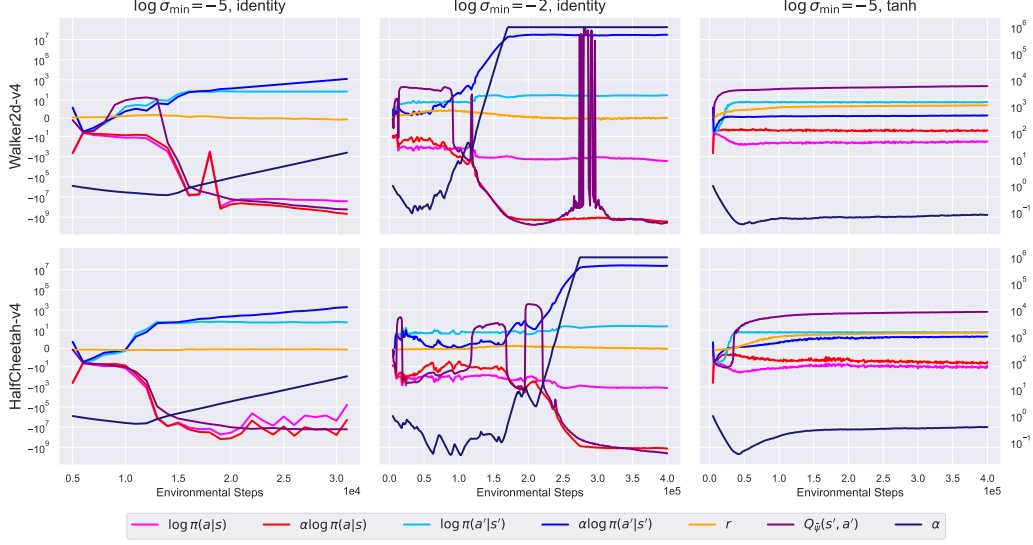

Figure 2: Scale comparison of the quantities in TD target. Left: $\log \sigma_{\min} = -5$, Middle: $\log \sigma_{\min} = -2$, Right: $\log \sigma_{\min} = -5$ with bounding by `tanh`. Top: `Walker2d-v4`, Bottom: `HalfCheetah-v4`. $\alpha$ is indicated by the right y-axis.

---

[1]More details on the setup and the metrics can be found in Section 5, and Figure 11 in Appendix B.2 shows the per-environment results.

We found that "bounding" $\alpha \log \pi_\theta$ terms improves the performance significantly. To be precise, by replacing the target $y(s, a, r, s', a')$ in the critic's loss (3) with the following, the agent succeeds to reach reasonable performance (the green learning curve in Figure 1; $\log \sigma_{\min} = -5$ is used):

$$y(s, a, r, s', a') = r + \beta \tanh\left(\alpha \log \pi_\theta(a|s)\right) + \gamma \left(Q_{\bar{\psi}}(s', a') - \tanh\left(\alpha \log \pi_\theta(a'|s')\right)\right). \tag{6}$$

The right column of Figure 2 shows that $Q_\psi$ is not ruined and $\alpha \log \pi_\theta$ terms do not explode. In the next section, we analyze what happens under the hood by theoretically investigating the effect of bounding $\alpha \log \pi_\theta$ terms. We argue that bounding $\alpha \log \pi_\theta$ terms is not just an ad-hoc implementation issue, but it changes the property of the underlying Bellman operator. We quantify the amount of ruin caused by $\alpha \log \pi_\theta$ terms, and show how this negative effect is mitigated by the bounding.

## 4 ANALYSIS

In this section, we theoretically investigate the properties of the log-policy-bounded target (6) in tabular settings. Rather than analyzing a specific choice of bounding, e.g. $\tanh(x)$, we characterize the conditions for bounding functions that are validated and effective. For the sake of analysis, we provide an abstract dynamic programming scheme of the log-policy-bounded target (6) and relate it to Advantage Learning (Baird, 1999; Bellemare et al., 2016) in Section 4.1. In Section 4.2, we show that carefully chosen bounding function ensures asymptotically convergence. In Section 4.3, we show that such bouding is indeed beneficial in terms of inherent error reduction property. All the proofs will be found in Appendix A.

### 4.1 BOUNDED ADVANTAGE LEARNING

Let $f$ and $g$ be non-decreasing functions over $\mathbb{R}$ such that, for both $h \in \{f, g\}$, (i) $h(x) > 0$ for $x > 0$, $h(x) < 0$ for $x < 0$ and $h(0) = 0$, (ii) $x - h(x) \geq 0$ for $x \geq 0$ and $x - h(x) \leq 0$ for $x \leq 0$, and (iii) their codomains are connected subsets of $[-c_h, c_h]$. The functions $\tanh(x)$ and $\text{clip}(x, -1, 1)$ satisfy these conditions. We understand that the identity map $I$ also satisfies these conditions with $c_h \to \infty$. Roughly speaking, we require the functions $f$ and $g$ to lie in the shaded area in Figure 3. Then, the loss (3), (4) and (6) can be seen as an instantiation of the following abstract VI scheme:

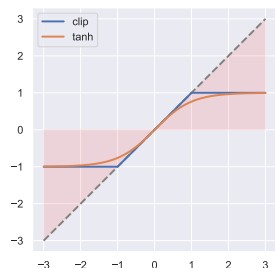

Figure 3: Examples of $f, g$.

$$\begin{cases} \pi_{k+1} = \mathcal{G}^{0,\alpha}(\Psi_k) \\ \Psi_{k+1} = R + \beta f\left(\alpha \log \pi_{k+1}\right) + \gamma P \left\langle \pi_{k+1}, \Psi_k - g\left(\alpha \log \pi_{k+1}\right) \right\rangle + \epsilon_{k+1} \end{cases}. \tag{7}$$

Notice that Munchausen-DQN and its variants are instantiations of this scheme, since their implementations clip the Munchausen bonus term by $f(x) = [x]_{l_0}^0$ with $l_0 = -1$ typically, while $g = I$. Furthermore, if we choose $f = g \equiv 0$, (7) reduces to Expected Sarsa (van Seijen et al., 2009).

Now, from the basic property of regularized MDPs, the soft state value function $V \in \mathbb{R}^{\mathcal{S}}$ satisfies $V = \alpha \log \left\langle \mu^\beta, \exp \frac{Q}{\alpha} \right\rangle = \alpha \log \left\langle \mathbf{1}, \exp \frac{\Psi}{\alpha} \right\rangle$, where $\Psi = Q + \beta \alpha \log \mu$. We write $\mathbb{L}^\alpha \Psi = \alpha \log \left\langle \mathbf{1}, \exp \frac{\Psi}{\alpha} \right\rangle$ for convention. The basic properties of $\mathbb{L}^\alpha$ are summarized in Appendix A.1. In the limit $\alpha \to 0$, it holds that $V(s) = \max_{a \in \mathcal{A}} \Psi(s, a)$. Furthermore, for a policy $\pi = \mathcal{G}^{0,\alpha}(\Psi)$, $\alpha \log \pi$ equals to the soft advantage function $A \in \mathbb{R}^{\mathcal{S} \times \mathcal{A}}$:

$$\alpha \log \pi = \alpha \log \frac{\exp \frac{\Psi}{\alpha}}{\left\langle \mathbf{1}, \exp \frac{\Psi}{\alpha} \right\rangle} = \alpha \log \exp \left(\frac{\Psi - V}{\alpha}\right) = \Psi - V =: A,$$

thus we have that $\alpha \log \pi_{k+1} = A_k$. Therefore, as discussed by Vieillard et al. (2020a), the recursion (2) is written as a soft variant of Advantage Learning (AL):

$$\Psi_{k+1} = R + \beta A_k + \gamma P \left\langle \pi_{k+1}, \Psi_k - A_k \right\rangle + \epsilon_{k+1} = R + \gamma P V_k - \beta(V_k - \Psi_k) + \epsilon_{k+1}.$$

Given these observations, we introduce a *bounded gap-increasing Bellman operator* $\mathcal{T}_{\pi_{k+1}}^{fg}$:

$$\mathcal{T}_{\pi_{k+1}}^{fg} \Psi_k = R + \beta f(A_k) + \gamma P \left\langle \pi_{k+1}, \Psi_k - g(A_k) \right\rangle. \tag{8}$$

Then, the DP scheme (7) is equivalent to the following *Bounded Advantage Learning* (BAL):

$$\begin{cases} \pi_{k+1} = \mathcal{G}^{0,\alpha}(\Psi_k) \\ \Psi_{k+1} = \mathcal{T}_{\pi_{k+1}}^{fg} \Psi_k + \epsilon_{k+1} \end{cases}. \tag{9}$$

By construction, the operator $\mathcal{T}_{\pi_{k+1}}^{fg}$ pushes-down the value of actions. To be precise, since $\max_{a \in \mathcal{A}} \Psi(s,a) \leq (\mathbb{L}^{\alpha}\Psi)(s)$, the soft advantage $A_k$ is always non-positive. Thus, the re-parameterized action value $\Psi_k$ is decreased by adding the term $\beta f(A_k)$. Obviously, the reduction is smallest at the optimal action $\arg\max_a \Psi_k(s,a)$. Therefore, the operator $\mathcal{T}_{\pi_{k+1}}^{fg}$ increases the action gaps with bounded magnitude dependent on $f$. In addition, as the term $-\gamma P \langle \pi_{k+1}, g(A_k) \rangle$ in Eq. (8) indicates, the entropy bonus for the successor state action pair $(s',a') \sim P_\pi(\cdot|s,a)$ is decreased by $g$.

We remark that BAL preserves the original mirror descent structure of MDVI (1). Noticing that $Q_k = \Psi_k - \beta\alpha \log \pi_k$, $(1-\beta)\alpha = \tau$ and $\beta\alpha = \lambda$, and following some steps similar to the derivation of Munchausen RL in Appendix A.2 of (Vieillard et al., 2020b), the bounded gap-increasing operator (8) can be rewritten in terms of $Q$ as

$$\tilde{\mathcal{T}}_{\pi_{k+1}|\pi_k}^{fg} Q_k = R - \beta \left( A_k - f(A_k) \right) + \gamma P \left( \langle \pi_{k+1}, Q_k + A_k - g(A_k) \rangle \right.$$
$$\left. + \tau \mathcal{H}(\pi_{k+1}) - \lambda D_{KL}(\pi_{k+1} \| \pi_k) \right).$$

Therefore, BAL still aligns the the original mirror descent structure of MDVI, but with additional modifications to the Bellman backup term. As we see later, the bounded gap-increasing operator (8) is more tolerant than AL and M-VI to *the errors of optimal policy misspecification*, which quantify the ruin caused by the soft advantage $A_k = \alpha \log \pi_{k+1}$.

## 4.2 CONVERGENCE OF BAL

First, we investigate the *asymptotic* converegnce property of BAL scheme. Since gap-increasing operators are *not contraction maps* in general, we need an argument similar to the analysis provided by Bellemare et al. (2016).

We start from the case where $\alpha \to 0$ while keeping $\beta$ constant, which corresponds to KL-only regularization. If an action-value function is updated using an operator $\mathcal{T}'$ that is *optimality-preserving*, at least one optimal action remains optimal, and suboptimal actions remain suboptimal. Further, if the operator $\mathcal{T}'$ is also *gap-increasing*, the value of suboptimal actions are pushed-down, which is advantageous in the presence of approximation or estimation errors (Farahmand, 2011) (please see Appendix A.2 for formal definitions). Notably, our operator $\mathcal{T}_{\pi_{k+1}}^{fg}$ is both optimality-preserving and gap-increasing in the limit $\alpha \to 0$.

**Theorem 1.** *In the limit $\alpha \to 0$, the operator $\mathcal{T}_{\pi_{k+1}}^{fg}$ satisfies $\mathcal{T}_{\pi_{k+1}}^{fg} \Psi_k \leq \mathcal{T}\Psi_k$ and $\mathcal{T}_{\pi_{k+1}}^{fg} \Psi_k \geq \mathcal{T}\Psi_k - \beta (V_k - \Psi_k)$ and thus is both optimality-preserving and gap-increasing.*

Next, we conisider the case $\alpha > 0$. The following theorem characterizes the possibly biased convergence of bounded gap-increasing operators under KL-entropy regularization.

**Theorem 2.** *Let $\Psi \in \mathbb{R}^{\mathcal{S} \times \mathcal{A}}$, $V = \mathbb{L}^\alpha \Psi$, $\mathcal{T}^\alpha \Psi = R + \gamma P \mathbb{L}^\alpha \Psi$ and $\mathcal{T}'$ be an operator with the properties that $\mathcal{T}' \Psi \leq \mathcal{T}^\alpha \Psi$ and $\mathcal{T}' \Psi \geq \mathcal{T}^\alpha \Psi - \beta (V - \Psi)$. Consider the sequence $\Psi_{k+1} := \mathcal{T}' \Psi_k$ with $\Psi_0 \in \mathbb{R}^{\mathcal{S} \times \mathcal{A}}$, and let $V_k = \mathbb{L}^\alpha \Psi_k$. Further, with an abuse of notation, we write $V_\tau^* \in \mathbb{R}^{\mathcal{S}}$ as the unique fixed point of the operator $\mathcal{T}^\tau V = \mathbb{L}^\tau (R + \gamma P V)$. Then, the sequence $(V_k)_{k \in \mathbb{N}}$ converges, and the limit $\tilde{V} = \lim_{k \to \infty} V_k$ satisfies $V_\tau^* \leq \tilde{V} \leq V_\alpha^*$. Furthermore, $\limsup_{k \to \infty} \Psi_k \leq Q_\alpha^*$ and $\liminf_{k \to \infty} \Psi_k \geq \frac{1}{1-\beta} \left( \tilde{Q} - \beta\tilde{V} \right)$, where $\tilde{Q} = R + \gamma P \tilde{V}$.*

Since $\mathcal{T}^\alpha \Psi_k \geq \mathcal{T}_{\pi_{k+1}}^{fI} \Psi_k = \mathcal{T}^\alpha \Psi_k + \beta f(A_k) \geq \mathcal{T}^\alpha \Psi_k + \beta A_k$, from Theorem 2 we can assure that BAL is convergent and $\Psi_k$ remains in a bounded range if $g = I$, even though $\tilde{V} \neq V_\tau^*$ in general. Furthermore, this result suggests that *Munchausen RL is convergent even when the ad-hoc clipping is employed*. However, Theorem 2 does not support the convergence for $g \neq I$, even though $g \neq I$ is empirically beneficial as seen in Section 3. The following Proposition 1 offers a sufficient condition for the asymptotic convergence when $g \neq I$, and characterizes the limiting behavior of BAL.

**Proposition 1.** *Consider the sequence* $\Psi_{k+1} := \mathcal{T}_{\pi_{k+1}}^{fg} \Psi_k$ *produced by the BAL operator* (8) *with* $\Psi_0 \in \mathbb{R}^{\mathcal{S} \times \mathcal{A}}$, *and let* $V_k = \mathbb{L}^\alpha \Psi_k$. *Assume that for all* $k \in \mathbb{N}$ *it holds that*

$$\lambda D_{\mathrm{KL}}(\pi_{k+1} \| \pi_k) - \gamma P^{\pi_{k+1}} \left( \alpha \mathcal{H}(\pi_{k+1}) + \langle \pi_{k+1}, g(A_k) \rangle \right) \geq 0. \tag{10}$$

*Then, the sequence* $(V_k)_{k \in \mathbb{N}}$ *converges, and the limit* $\tilde{V} = \lim_{k \to \infty} V_k$ *satisfies* $V_\tau^* - \frac{\gamma}{1-\gamma} \frac{\alpha}{1-\beta} \log |\mathcal{A}| \leq \tilde{V} \leq V_\alpha^*$. *Furthermore,* $\limsup_{k \to \infty} \Psi_k \leq Q_\alpha^*$ *and* $\liminf_{k \to \infty} \Psi_k \geq \frac{1}{1-\beta} \left( \tilde{Q} - \beta \tilde{V} - \gamma \alpha \log |\mathcal{A}| \right)$, *where* $\tilde{Q} = R + \gamma P \tilde{V}$.

We remark that the lower bound $V_\tau^* - \frac{\gamma}{1-\gamma} \frac{\alpha}{1-\beta} \log |\mathcal{A}|$ makes sense. Since $V_{\max}^\tau = V_{\max} + \frac{\tau \log |\mathcal{A}|}{1-\gamma}$, the magnitude of the lower bound roughly matches the un-regularized value, which appears because $g$ decreases the entropy bonus in the Bellman backup. One way to satisfy (10) for all $k \in \mathbb{N}$ is to use an adaptive strategy to determine $g$. Since $\pi_{k+1}$ is obtained *before* the update $\Psi_{k+1} = \mathcal{T}_{\pi_{k+1}}^{fg} \Psi_k$ in BAL scheme (9), it is possible that we first compute $D_{\mathrm{KL}}(\pi_{k+1} \| \pi_k)$ and $\mathcal{H}(\pi_{k+1})$, and then adaptively find $g$ that satisfies (10), with additional computational efforts. In the following, however, we provide an error propagation analysis and argue that a fixed $g \neq I$ is indeed beneficial.

## 4.3 BOUNDING DECREASES THE ERRORS OF OPTIMAL POLICY MISSPECIFICATION

Theorem 2 indicates that BAL is convergent but possibly biased even when $g = I$. However, we can still upper-bound the error between the optimal entropy-regularized state value $V_\tau^*$, which is the unique fixed point of the operator $\mathcal{T}^\tau V = \mathbb{L}^\tau (R + \gamma P V)$, and the entropy-regularized state value $V_\tau^{\pi_k}$ for the sequence of the policies $\{\pi_k\}_k$ generated by BAL. Theorem 3 below, which generalizes Theorem 1 in Zhang et al. (2022) to KL-entropy-regularized settings with the bounding functions $f$ and $g$, provides such a bound and highlights the advantage of BAL for both $f \neq I$ and $g \neq I$.

**Theorem 3.** *Let* $\{\pi_k\}_k$ *be a sequence of the policies obtained by BAL. Defining* $\Delta_k^{fg} = \langle \pi^*, \beta \left( A_\tau^* - f(A_{k-1}) \right) - \gamma P \langle \pi_k, A_{k-1} - g(A_{k-1}) \rangle \rangle$, *it holds that:*

$$\| V_\tau^* - V_\tau^{\pi_{K+1}} \|_\infty \leq \frac{2\gamma}{1-\gamma} \left[ 2\gamma^{K-1} V_{\max}^\tau + \sum_{k=1}^{K-1} \gamma^{K-k-1} \left\| \Delta_k^{fg} \right\|_\infty \right]. \tag{11}$$

Since the suboptimality of BAL is characterize by Theorem 3, we can discuss its convergence property as in previous researches (Kozuno et al., 2019; Vieillard et al., 2020a). The bound (11) resembles the standard suboptimality bounds in the literature (Munos, 2005; 2007; Antos et al., 2008; Farahmand et al., 2010), which consists of the horizon term $2\gamma/(1-\gamma)$, initialization error $2\gamma^{K-1} V_{\max}^\tau$ that goes to zero as $K \to \infty$, and the accumulated error term. However, our error terms do not represent the Bellman backup errors, but capture *the misspecifications of the optimal policy* as we discuss later. We note that, the error term $\Delta_k^{fg}$ does not contain the error $\epsilon_k$, because we simply omitted it in our analysis as done by Zhang et al. (2022). Our interest here is *not* in the effect of the approximation/estimation error $\epsilon_k$, but in the effect of *the ruin caused by the soft advantage* $A_k = \alpha \log \pi_{k+1}$, that is, the error inherent to the soft-gap-increasing nature of M-VI and BAL in model-based tabular settings without any approximation. In the following, we consider a decompostion of the error $\Delta_k^{fg} = \Delta_k^{Xf} + \Delta_k^{Hg}$ and argue that (1) the cross term $\Delta_k^{Xf} = -\beta \langle \pi^*, f(A_{k-1}) \rangle$ has major effect on the sub-optimality and is *always* decreased by $f \neq I$, and (2) the entropy terms $\Delta_k^{Hg} = \langle \pi^*, \beta A_\tau^* - \gamma P \langle \pi_k, A_{k-1} - g(A_{k-1}) \rangle \rangle$ are decreased by $g \neq I$, although which is *not always* true.

To ease the exposition, first let us again consider the case $\alpha \to 0$ while keeping $\beta > 0$ constant. Then, noticing that we have $\mathcal{G}^{0,0}(\Psi) = \mathcal{G}(\Psi)$, $\mathbb{L}^\alpha \Psi(s) \to \max_{b \in \mathcal{A}} \Psi(s, b)$ and $g(0) = 0$, it follows that the entropy terms are equal to zero: $\langle \pi^*, A^* \rangle = \langle \pi_{k+1}, A_k \rangle = \langle \pi_{k+1}, g(A_k) \rangle = 0$. Thus, $\Delta_k^{fg}$ reduces to $\Delta_k^{Xf} = -\beta \langle \pi^*, f(A_{k-1}) \rangle$ and $\Delta_k^{Xf}(s) = -\beta f \left( \Psi_{k-1}(s, \pi^*(s)) - \Psi_{k-1}(s, \pi_k(s)) \right)$. Therefore, $\Delta_k$ represents *the error incurred by the misspecification of the optimal policy*. For AL, the error is $\Delta_k^{XI}(s) = \beta \left( \Psi_{k-1}(s, \pi_k(s)) - \Psi_{k-1}(s, \pi^*(s)) \right)$. Since both AL and BAL are optimality-preserving for $\alpha \to 0$, we have $\| \Delta_k^{XI} \|_\infty \to 0$ and $\| \Delta_k^{Xf} \|_\infty \to 0$ as $k \to \infty$. Howerver, their convergence speed is governed by the magnitude of $\| \Delta_k^{XI} \|_\infty$ and $\| \Delta_k^{Xf} \|_\infty$ at finite $k$, respectively.

We remark that for all $k$ it holds that $|\Delta_k^{\mathrm{X}f}| \leq |\Delta_k^{\mathrm{X}I}|$ point-wise. Indeed, from the non-positivity of $A_k$ and the requirement to $f$, we always have $A_k = I(A_k) \leq f(A_k)$ point-wise and then $-\beta I(A_k(s,a)) \geq -\beta f(A_k(s,a))$ for all $(s,a)$ and $k$, both sides of which are non-negative. Thus, we have $\langle \pi^*, -\beta f(A_{k-1}) \rangle \leq \langle \pi^*, -\beta I(A_{k-1}) \rangle$ point-wise and therefore $|\Delta_k^{\mathrm{X}f}| \leq |\Delta_k^{\mathrm{X}I}|$. Furthermore, we have $\|\Delta_k^{\mathrm{X}I}\|_\infty \leq \frac{2R_{\max}}{1-\gamma}$ for AL while $\|\Delta_k^{\mathrm{X}f}\|_\infty \leq c_f$ for BAL. Therefore, BAL has better convergence property than AL by a factor of the horizon $1/(1-\gamma)$ in the case where $\Psi_k$ is far from optimal.

For the case $\alpha > 0$, $\|\Delta_k^{fg}\|_\infty \to 0$ does not hold in general. Further, the entropy terms are no longer equal to zero. However, the cross term, which is an order of $1/(1-\gamma)$, is much larger unless the action space is extremely large since the entropy is an order of $\log|\mathcal{A}|$ at most, and is always decreased by $f \neq I$. Furthermore, we can expect that $g \neq I$ decreases the error $\Delta_k^{\mathcal{H}g}$, though it does *not always* true. If $g \neq I$, the entropy terms reduce to $\Delta_k^{\mathcal{H}I} = \langle \pi^*, \beta A^* \rangle$. Since $A_{k-1}$ is non-positive, we have $A_{k-1} - g(A_{k-1}) \leq 0$ from the requirements to $g$. Since the stochastic matrix $P$ is non-negative, we have $P \langle \pi_k, A_{k-1} - g(A_{k-1}) \rangle \leq 0$, where the l.h.s. represents the decreased negative entropy of the successor state and its absolute value is again an order of $\log|\mathcal{A}|$ at most. Since $A^* \leq 0$ also, whose absolute value is an order of $1/(1-\gamma)$, it holds that $\beta A^* \leq \beta A^* - \gamma P \langle \pi_k, A_{k-1} - g(A_{k-1}) \rangle$ and thus $\Delta_k^{\mathcal{H}I} = \langle \pi^*, \beta A^* \rangle \leq \langle \pi^*, \beta A_\tau^* - \gamma P \langle \pi_k, A_{k-1} - g(A_{k-1}) \rangle \rangle = \Delta_k^{\mathcal{H}g}$. When $\Delta_k^{\mathcal{H}g}$ is non-positive, it is guaranteed that $|\Delta_k^{\mathcal{H}g}| \leq |\Delta_k^{\mathcal{H}I}|$. In addition, we can expect that this error is largely decreased by zero function $g(x) \equiv 0$, though it makes harder to satisfy the inequality (10). However, this inequality does not always hold because it depends on the actual magnitude of $A^*$ and $P \langle \pi_k, A_{k-1} - g(A_{k-1}) \rangle$.

Overall, there is a trade-off in the choice of $g$; $g = I$ always satisfies the sufficient condition of asymptotic convergence (10), but the entropy term is not decreased. On the other hand, $g(x) \equiv 0$ is expected to decrease the entropy term, though which possibly violates (10) and might hinder the asymptotic performance. In the next section, we examine how the choice of $f$ and $g$ affects the empirical performance.

## 5 EXPERIMENT

### 5.1 BAL ON GRID WORLD

First, we compare the model-based tabular M-VI (2) and BAL (9) schemes. As discussed by Vieillard et al. (2020a), the larger the value of $\beta$ is, the slower the initial convergence of MDVI gets, and thus M-VI as well. Since the reduction of the misspecification error by BAL is particularly effective when $\Psi_k$ is far from the optimal, we can expect that BAL is effective especially in earlier iterations. We vaidate this hypothesis by a model-based tabular setting.

We use a gridworld environment, where transition kernel $P$ and reward function $R$ are directly available. We performed 100 independent runs with random initialization of $\Psi_0$. Figure 4 compares the normalized value of the suboptimality $\|V^{\pi_k} - V_\tau^*\|_\infty$, where the interquatile mean (IQM) is reported as suggested by Agarwal et al. (2021). The result suggests that BAL outperforms M-VI initially. Furthermore, $g \neq I$ performs slightly better than $g = I$ in the earlier stage, even in this toy problem. Therefore, it is validated that BAL is effective especially in earlier iterations. More experimental details are found in Appendix B.1.

### 5.2 MDAC ON MUJOCO LOCOMOATION ENVIRONMENTS

**Setup and Metrics.** Next, we empirically evaluate the effectiveness of MDAC on 6 Mujoco environments (`Hopper-v4`, `HalfCheetah-v4`, `Walker2d-v4`, `Ant-v4`, `Humanoid-v4` and `HumanoidStandup-v4`) from Gymnasium (Towers et al., 2023). We evaluate our algorithm and baselines on 3M environmental steps, except for easier `Hopper-v4` on 1M steps. For the reliable benchmarking, we again report the aggregated scores over all 6 environments as suggested by Agarwal et al. (2021). To be precise, we train 10 different instances of each algorithm with different random seeds and calculate baseline-normalized scores along iterations for each task as score $= \frac{\mathrm{score}_{\mathrm{algorithm}} - \mathrm{score}_{\mathrm{random}}}{\mathrm{score}_{\mathrm{baseline}} - \mathrm{score}_{\mathrm{random}}}$, where the baseline is the mean SAC score after 3M steps (1M for `Hopper-v4`). Then, we calculate the IQM score by aggregating the learning results over all

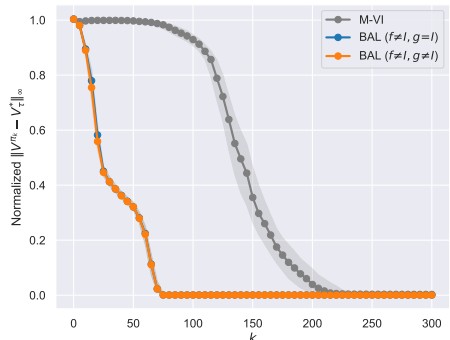

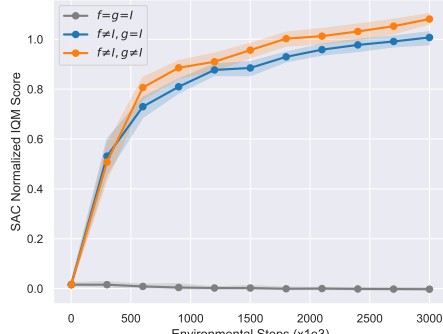

Figure 4: Results of M-VI and BAL on Gridworld.   Figure 5: Effect of $f \neq I$ and $g \neq I$ on Mujoco.

6 environments. We also report pointwise 95% percentile stratified bootstrap confidence intervals. We use Adam optimizer (Kingma & Ba, 2015) for all the gradient-based updates. The discount factor is set to $\gamma = 0.99$. All the function approximators, including those for baseline algorithms, are fully-connected feed-forward networks with two hidden layers and each hidden layer has 256 units with ReLU activations. We use a Gaussian policy with mean and standard deviation provided by the neural network. We fixed $\log \sigma_{\min} = -5$. More experimental details, including a full list of the hyperparameters and per-environment results, will be found in Appendix B.2.

**Effect of bounding functions $f$ and $g$.**   We start from evaluating how the performance of MDAC is affected by the choice of the bounding functions. First, we evaluate whether bounding both $\log \pi(a|s)$ terms is beneficial. We compare 3 choices: (i) $f = g = I$, (ii) $f(x) = \tanh(x/10), g = I$ and (iii) $f(x) = g(x) = \tanh(x/10)$. Figure 5 compares the learning results for these choices and it indicates that bounding both $\alpha \log \pi$ terms is indeed beneficial.

Next, we compare 5 choices under $f = g \neq I$: (i) clip$(x, -1, 1)$, (ii) clip$(x/10, -1, 1)$, (iii) $\tanh(x)$, (iv) $\tanh(x/10)$, and (v) sign$(x)$. Notice that the last choice (v) violates our requirement to the bounding functions. Figure 6 compares the learning curves for these choices. The result indicates that the performence difference between clip$(x)$ and $\tanh(x)$ is small. On the other hand, the performance is boosted if the slower saturating functions are used. Furthermore, sign$(x)$ resulted in the worst performance among these choices. Figure 7 compares the frequencies of clipping $\alpha \log \pi$ terms by clip$(x, -1, 1)$ and clip$(x/10, -1, 1)$ in the sampled minibatchs for the

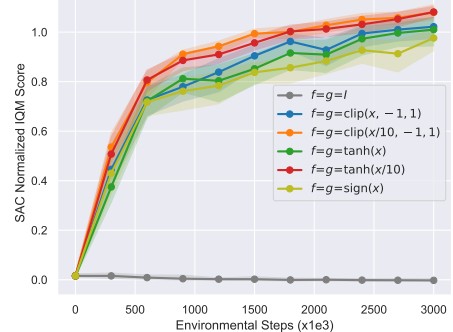

Figure 6: Comparison of $f$ and $g$.

initial learning phase in `Walker2d-v4`, `HalfCheetah-v4` and `Ant-v4`. For clip$(x, -1, 1)$, the clipping occurs frequently especially for the current $(s, a)$ pairs and the information of relative $\alpha \log \pi$ values between different state-actions are lost. On the other hand, for clip$(x/10, -1, 1)$, the clipping rarely happens and the information of relative $\alpha \log \pi$ values are leveraged in the learning. These results suggest that the relative values of $\alpha \log \pi$ tems between different state-actions are beneficial for the learning process, even though the raw values (by $f = g = I$) are harmful.

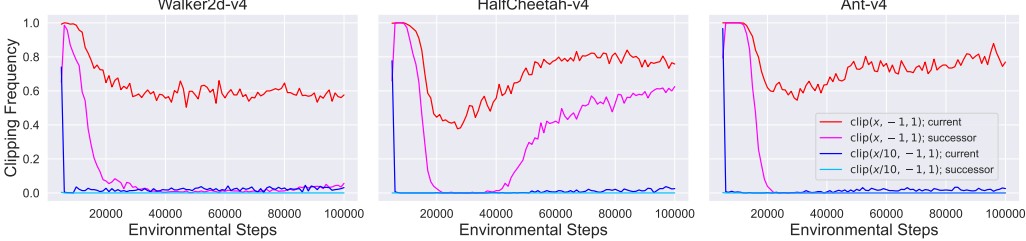

Figure 7: Comparison of clipping frequencies by $f(x) = g(x) = $ clip$(x, -1, 1)$ and $f(x) = g(x) = $ clip$(x/10, -1, 1)$ in early learning stage.

**Comparison to baseline algorithms.** We compare MDAC against SAC (Haarnoja et al., 2018b), an entropy-only-regularized method, and TD3 (Fujimoto et al., 2018), a non-regularized method. We adopted $f(x) = g(x) = \text{clip}(x/10, -1, 1)$. Figure 8 compares the learning results. Notice that the final IQM score of SAC does not match 1, because the scores are normalized by the mean of all the SAC runs, whereas IQM is calculated by middle 50% runs. The results show that MDAC overtakes both SAC and TD3. Roughly speaking, MDAC requires only the half amount of samples to reach reasonable performance compared to SAC.

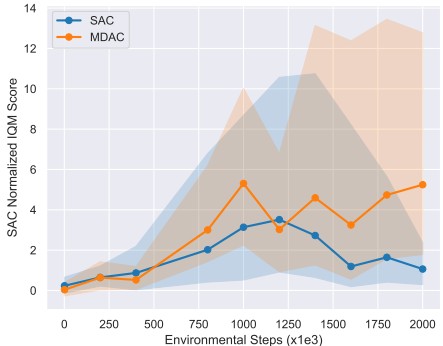

Figure 8: Benchmarking results on Mujoco.

### 5.3 MDAC ON DEEPMIND CONTROL SUITE

Finally, we compare MDAC and SAC on challenging `dog` domain from DeepMind Control Suite (Tunyasuvunakool et al., 2020). We adopted `stand`, `walk`, `trot` and `run` tasks. We train 10 different instances of each algorithm for 2M environmental steps, and report SAC normalized IQM scores. We adopted $f(x) = g(x) = \text{clip}(x/10, -1, 1)$ for MDAC again. Hyperparameters are set to equivalent values as Mujoco experiments. Figure 9 compares the learning results. Though the aggregated result is not statistically strong, MDAC tends to reach better performace than SAC especially in `walk` and `run`. While the performances of both algorithms often degrade during the learning due to the difficulty of the `dog` domain, this degradation is slightly mild for MDAC. We conjecture that this effect is due to the implicit KL-regularized nature of MDAC.

Figure 9: Learning results on DeepMind Control Suite `dog` envoironments.

## 6 CONCLUSION

In this study, we proposed MDAC, a model-free actor-critic instantiation of MDVI for continuous action domains. We showed that its empirical performance is significantly boosted by bounding the values of log-density terms in the critic loss. By relating MDAC to AL, we theoretically showed that the error of optimal policy misspecification is decreased by bounding the advantage terms, as well as the convergence analyses. Our analysis indicated that bounding both of the log-policy terms is beneficial. Lastly, we evaluated the effectiveness of MDAC empirically in simulated environments.

**Limitations.** This study has three major limitations. First, our theoretical analyses are valid only for fixed $\alpha$. Thus, its exploding behavior observed in Section 3 for $f = g = I$ is not captured. Second, our theoretical analyses apply only to tabular cases in the current forms. To extend our analyses to continuous state-action domains, we need measure-theoretic considerations as explored in Appendix B of (Puterman, 1994). Last, our analyses and experiments do not offer the optimal design of the bounding functions $f$ and $g$. We leave these issues as open questions.

**Ethics Statement.** Although the work presented here has an academic nature mostly, it helps the development of capable autonomous agents. While our contributions do not have a direct path to negative societal impacts, we urge that these must be considered when our research is applied.

**Reproducibility Statement.** The proofs for our theoretical results are formaly provided in Appendix A. Our theoretical statements include their assumptions. Please be noticed that the focus of our paper is limited to MDP. Regarding the experimental reproducibility; we submitted the anonymized code to reproduce our experimental results. We provided the essential information of experimental settings in Section 5. We also provided further experimental details in Appendix B.

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

# A PROOFS

## A.1 BASIC PROPERTIES OF $\mathbb{L}^\alpha$

In this section, we omit $\Psi$'s dependency to state $s$ for the brevity. Let $\Psi \in \mathbb{R}^{\mathcal{A}}$. For $\alpha > 0$, we write $\mathbb{L}^\alpha \Psi = \alpha \log \left\langle \mathbf{1}, \exp \frac{\Psi}{\alpha} \right\rangle \in \mathbb{R}$.

**Lemma 1.** *It holds that*

$$\max_{a \in \mathcal{A}} \Psi(a) \leq \mathbb{L}^\alpha \Psi \leq \max_{a \in \mathcal{A}} \Psi(a) + \alpha \log |\mathcal{A}|.$$

*Proof.* Let $y = \max_{a \in \mathcal{A}} \Psi(a)$. We have that

$$\exp \frac{y}{\alpha} \leq \left\langle \mathbf{1}, \exp \frac{\Psi}{\alpha} \right\rangle = \sum_{a \in \mathcal{A}} \exp \frac{\Psi(a)}{\alpha} \leq |\mathcal{A}| \exp \frac{y}{\alpha}.$$

Applying the logarithm to this inequality, we have

$$\frac{y}{\alpha} \leq \log \left\langle \mathbf{1}, \exp \frac{\Psi}{\alpha} \right\rangle \leq \frac{y}{\alpha} + \log |\mathcal{A}|,$$

and thus the claim follows. ∎

**Lemma 2.** *It holds that* $\lim_{\alpha \to 0} \mathbb{L}^\alpha \Psi \to \max_{a \in \mathcal{A}} \Psi(a)$.

*Proof.* Let $y = \max_{a \in \mathcal{A}} \Psi(a)$ and $\mathcal{B} = \{a \in \mathcal{A} | \Psi(a) = y\}$. It holds that

$$\begin{aligned}
\lim_{\alpha \to 0} \mathbb{L}^\alpha \Psi &= \lim_{\alpha \to 0} \alpha \log \sum_{a \in \mathcal{A}} \exp \frac{\Psi(a)}{\alpha} \\
&= \lim_{\alpha \to 0} \alpha \log \left( \exp \frac{y}{\alpha} \sum_{a \in \mathcal{A}} \exp \frac{\Psi(a) - y}{\alpha} \right) \\
&= y + \lim_{\alpha \to 0} \alpha \log \left( \sum_{a \in \mathcal{B}} \underbrace{\exp \frac{\Psi(a) - y}{\alpha}}_{=1} + \sum_{a \notin \mathcal{B}} \exp \frac{\Psi(a) - y}{\alpha} \right) \\
&= y + \lim_{\alpha \to 0} \alpha \log \left( |\mathcal{B}| + \sum_{a \notin \mathcal{B}} \exp \frac{\Psi(a) - y}{\alpha} \right).
\end{aligned}$$

Since $\Psi(a) - y < 0$ for $a \in \mathcal{B}$, we have $\exp \frac{\Psi(a) - y}{\alpha} \to 0$ for $a \in \mathcal{B}$, which concludes the proof. ∎

## A.2 PROOF OF THEOREM 1

We start from providing the formal definition of *optimality-preserving* and *gap-increasing*.

**Definition 1** (Optimality-preserving). *An operator* $\mathcal{T}'$ *is optimality-preserving if, for any* $Q_0 \in \mathbb{R}^{\mathcal{S} \times \mathcal{A}}$ *and* $s \in \mathcal{S}$, *letting* $Q_{k+1} := \mathcal{T}' Q_k$, $\tilde{V}(s) := \lim_{k \to \infty} \max_{b \in \mathcal{A}} Q_k(s, b)$ *exists, is unique,* $\tilde{V}(s) = V^*(s)$, *and for all* $a \in \mathcal{A}$, $Q^*(s, a) < V^*(s, a) \implies \limsup_{k \to \infty} Q_k(s, a) < V^*(s)$.

**Definition 2** (Gap-increasing). *An operator* $\mathcal{T}'$ *is gap-increasing if for all* $Q_0 \in \mathbb{R}^{\mathcal{S} \times \mathcal{A}}$, $s \in \mathcal{S}, a \in \mathcal{A}$, *letting* $Q_{k+1} := \mathcal{T}' Q_k$ *and* $V_k(x) := \max_b Q_k(s, b)$, $\liminf_{k \to \infty} \left[ V_k(s) - Q_k(s, a) \right] \geq V^*(s) - Q^*(s, a)$.

The following lemma characterizes when an operator is optimality-preserving and gap-increasing.

**Lemma 3** (Theorem 1 in (Bellemare et al., 2016)). *Let* $V(s) := \max_b Q(s, b)$ *and let* $\mathcal{T}$ *be the Bellman optimality operator* $\mathcal{T} Q = R + \gamma P V$. *Let* $\mathcal{T}'$ *be an operator with the property that there exists an* $\rho \in [0, 1)$ *such that for all* $Q \in \mathbb{R}^{\mathcal{S} \times \mathcal{A}}$, $s \in \mathcal{S}, a \in \mathcal{A}$, $\mathcal{T}' Q \leq \mathcal{T} Q$, *and* $\mathcal{T}' Q \geq \mathcal{T} Q - \rho (V - Q)$. *Then* $\mathcal{T}'$ *is both optimality-preserving and gap-increasing.*

Now, we state our Theorem 1 again.

**Theorem 4** (Theorem 1 in the main text)**.** *In the limit $\alpha \to 0$, the operator $\mathcal{T}^{fg}_{\pi_{k+1}}$ satisfies $\mathcal{T}^{fg}_{\pi_{k+1}} \Psi_k \leq \mathcal{T} \Psi_k$ and $\mathcal{T}^{fg}_{\pi_{k+1}} \Psi_k \geq \mathcal{T} \Psi_k - \beta (V_k - \Psi_k)$ and thus is both optimality-preserving and gap-increasing.*

*Proof.* From Lemma 2, we have $\mathbb{L}^\alpha(s)\Psi \to \max_{a \in \mathcal{A}} \Psi(s, a)$ as $\alpha \to 0$ for $\Psi \in \mathbb{R}^{\mathcal{S} \times \mathcal{A}}$. Observe that, for $h \in \{f, g\}$, it holds that $h(A_k) = h(\Psi_k - V_k) \leq 0$ since $A_k(s, a) = \Psi_k(s, a) - \max_{b \in \mathcal{A}} \Psi_k(s, b) \leq 0$ and $h$ does not flip the sign of argument. Additionally, for $\pi_{k+1} \in \mathcal{G}(\Psi_k)$ it follows that $\langle \pi_{k+1}, h(A_k) \rangle = 0$ since $h(0) = 0$. It holds that

$$\mathcal{T}^{fg}_{\pi_{k+1}} \Psi_k - \mathcal{T} \Psi_k = R + \beta f(A_k) + \gamma P \langle \pi_{k+1}, \Psi_k - g(A_k) \rangle - R - \gamma P \langle \pi_{k+1}, \Psi_k \rangle$$
$$= \beta \underbrace{f(A_k)}_{\leq 0} - \gamma P \underbrace{\langle \pi_{k+1}, g(A_k) \rangle}_{=0} \leq 0.$$

Furthermore, observing that $x - f(x) \leq 0$ for $x \leq 0$, it follows that

$$\mathcal{T}^{fg}_{\pi_{k+1}} \Psi_k - \mathcal{T} \Psi_k + \beta (V_k - \Psi_k) = -\beta \underbrace{(A_k - f(A_k))}_{\leq 0} - \gamma P \underbrace{\langle \pi_{k+1}, g(A_k) \rangle}_{=0} \geq 0.$$

Thus, the operator $\mathcal{T}^{fg}_{\pi_{k+1}}$ satisfies the conditions of Lemma 3. Therefore we conclude that $\mathcal{T}^{fg}_{\pi_{k+1}}$ is both optimality-preserving and gap-increasing. ∎

### A.3 PROOF OF THEOREM 2

We provide several lemmas that are used to prove Theorem 2.

**Lemma 4.** *For $Q \in \mathbb{R}^{\mathcal{S} \times \mathcal{A}}$, let $V = \mathbb{L}^\tau Q$ and $\Psi' = \frac{Q - \beta V}{1 - \beta}$. Then it holds that $\mathbb{L}^\alpha \Psi' = V$.*

*Proof.* It holds that

$$\mathbb{L}^\alpha \Psi' = \alpha \log \left\langle \mathbf{1}, \exp \frac{1}{\alpha} \frac{Q - \beta V}{1 - \beta} \right\rangle$$
$$= \alpha \log \left\langle \mathbf{1}, \exp \left( \frac{1}{\alpha} \frac{Q}{1 - \beta} \right) \right\rangle + \alpha \log \exp \left( -\frac{1}{\alpha} \frac{\beta V}{1 - \beta} \right)$$
$$= \mathbb{L}^\alpha \frac{Q}{1 - \beta} - \frac{\beta V}{1 - \beta}.$$

We have

$$\mathcal{G}^{0,\alpha} \left( \frac{Q}{1 - \beta} \right) = \frac{\exp \frac{1}{\alpha} \frac{Q}{1 - \beta}}{\left\langle \mathbf{1}, \exp \frac{1}{\alpha} \frac{Q}{1 - \beta} \right\rangle} = \frac{\exp \frac{Q}{\tau}}{\left\langle \mathbf{1}, \exp \frac{Q}{\tau} \right\rangle} = \mathcal{G}^{0,\tau} (Q) =: \pi_\tau,$$

and thus

$$\mathbb{L}^\alpha \frac{Q}{1 - \beta} = \left\langle \pi_\tau, \frac{Q}{1 - \beta} \right\rangle + \alpha \mathcal{H}(\pi_\tau) = \frac{1}{1 - \beta} \left( \langle \pi_\tau, Q \rangle + (1 - \beta)\alpha \mathcal{H}(\pi_\tau) \right) = \frac{1}{1 - \beta} \mathbb{L}^\tau Q.$$

Thus it follows that $\mathbb{L}^\alpha \Psi' = V$. ∎

**Lemma 5.** *Let $\Psi \in \mathbb{R}^{\mathcal{S} \times \mathcal{A}}$, $V = \mathbb{L}^\alpha \Psi$ and $\mathcal{T}'$ be an operator with the properties that $\mathcal{T}' \Psi \leq \mathcal{T}^\alpha \Psi$ and $\mathcal{T}' \Psi \geq \mathcal{T}^\alpha \Psi - \beta (V - \Psi) = \mathcal{T}^\alpha \Psi + \beta (A)$. Consider the sequence $\Psi_{k+1} := \mathcal{T}' \Psi_k$ with $\Psi_0 \in \mathbb{R}^{\mathcal{S} \times \mathcal{A}}$, and let $V_k = \mathbb{L}^\alpha \Psi_k$. Then the sequence $(V_k)_{k \in \mathbb{N}}$ converges.*

*Proof.* From $\mathcal{T}' \Psi \leq \mathcal{T}^\alpha \Psi$ and observing that $\mathcal{T}^\alpha$ has a unique fixed point, we have

$$\limsup_{k \to \infty} \Psi_k = \limsup_{k \to \infty} (\mathcal{T}')^k \Psi_0 \leq \limsup_{k \to \infty} (\mathcal{T}^\alpha)^k \Psi_0 = Q^*_\alpha. \tag{12}$$

Thus, $\limsup_{k\to\infty} \Psi_k =: \tilde{\Psi}$ is upper-bounded. Let $\tilde{V} := \limsup_{k\to\infty} V_k$. We will see that $\liminf_{k\to\infty} V_k = \tilde{V}$ also. We have

$$
\begin{aligned}
V_{k+1} = \mathbb{L}^\alpha \Psi_{k+1} &= \langle \pi_{k+2}, \Psi_{k+1} \rangle + \alpha \mathcal{H}(\pi_{k+2}) \\
&\geq \langle \pi_{k+1}, \Psi_{k+1} \rangle + \alpha \mathcal{H}(\pi_{k+1}) \\
&= \langle \pi_{k+1}, \mathcal{T}' \Psi_k \rangle + \alpha \mathcal{H}(\pi_{k+1}) \\
&\geq \langle \pi_{k+1}, \mathcal{T}^\alpha \Psi_k + \beta A_k \rangle + \alpha \mathcal{H}(\pi_{k+1}) \\
&\overset{(a)}{=} \langle \pi_{k+1}, \mathcal{T}^\alpha \Psi_k \rangle + (1 - \beta)\alpha \mathcal{H}(\pi_{k+1}) \\
&\overset{(b)}{=} \langle \pi_{k+1}, Q_k + \gamma P(V_k - V_{k-1}) \rangle + (1 - \beta)\alpha \mathcal{H}(\pi_{k+1}) \\
&\overset{(c)}{=} \langle \pi_{k+1}, Q_k + \gamma P(V_k - V_{k-1}) \rangle + \tau \mathcal{H}(\pi_{k+1}) - \lambda D_{\mathrm{KL}}(\pi_{k+1}\|\pi_k) + \lambda D_{\mathrm{KL}}(\pi_{k+1}\|\pi_k) \\
&\overset{(d)}{=} V_k + \langle \pi_{k+1}, \gamma P(V_k - V_{k-1}) \rangle + \lambda D_{\mathrm{KL}}(\pi_{k+1}\|\pi_k) \\
&\geq V_k + \langle \pi_{k+1}, \gamma P(V_k - V_{k-1}) \rangle,
\end{aligned}
$$

where (a) follows from $\langle \pi_{k+1}, A_k \rangle = \langle \pi_{k+1}, \alpha \log \pi_{k+1} \rangle = -\alpha \mathcal{H}(\pi_{k+1})$, (b) follows from $\mathcal{T}^\alpha \Psi_k = R + \gamma P \mathbb{L}^\alpha \Psi_k = R + \gamma P V_k = Q_{k+1}$, (c) follows from $(1 - \beta)\alpha = \tau$, and (d) follows from $V_k = \mathbb{L}^\alpha \Psi_k = \langle \pi_{k+1}, Q_k \rangle + \tau \mathcal{H}(\pi_{k+1}) - \lambda D_{\mathrm{KL}}(\pi_{k+1}\|\pi_k)$. Thus we have

$$
V_{k+1} - V_k \geq \gamma P^{\pi_{k+1}}(V_k - V_{k-1})
$$

and by induction

$$
V_{k+1} - V_k \geq \gamma^k P_{k+1:2}(V_1 - V_0),
$$

where $P_{k+1:2} = P^{\pi_{k+1}} P^{\pi_k} \cdots P^{\pi_2}$. From the conditions on $\mathcal{T}'$, if $V_0$ is bounded then $V_1$ is also bounded, and thus $\|V_1 - V_0\|_\infty < \infty$. By definition, for any $\delta > 0$ and $n \in \mathbb{N}$, $\exists k \geq n$ such that $V_k > \tilde{V} - \delta$. Since $P_{k+1:2}$ is a nonexpansion in $\infty$-norm, we have

$$
V_{k+1} - V_k \geq -\gamma^k \|V_1 - V_0\|_\infty \geq -\gamma^n \|V_1 - V_0\|_\infty =: -\epsilon,
$$

and for all $t \in \mathbb{N}$,

$$
V_{k+t} - V_k \geq -\sum_{i=0}^{t-1} \gamma^i \epsilon \geq \frac{-\epsilon}{1 - \gamma}.
$$

Thus, we have

$$
\inf_{t \in \mathbb{N}} V_{k+t} \geq V_k - \frac{\epsilon}{1 - \gamma} > \tilde{V} - \delta - \frac{\epsilon}{1 - \gamma}.
$$

It follows that for any $\delta' > 0$, we can choose an $n \in \mathbb{N}$ to make $\epsilon$ small enough such that for all $k \geq n$, $V_k > \tilde{V} - \delta'$. Hence

$$
\liminf_{k\to\infty} V_k = \tilde{V},
$$

and thus $V_k$ converges. ∎

**Lemma 6.** *Let $\mathcal{T}'$ be an operator satisfying the conditions of Lemma 5. Then for all $k \in \mathbb{N}$,*

$$
|V_k| \leq \frac{1}{1 - \gamma}\Big[3\|V_0\|_\infty + R_{\max} + \alpha \log|\mathcal{A}|\Big]. \tag{13}
$$

*Proof.* Following the derivation of Lemma 5, we have

$$
V_{k+1} - V_0 \geq -\sum_{i=1}^{k} \gamma^i \|V_1 - V_0\|_\infty \geq \frac{-1}{1 - \gamma}\|V_1 - V_0\|_\infty. \tag{14}
$$

We also have

$$
V_1 = \mathbb{L}^\alpha \mathcal{T}' \Psi_0 \leq \mathbb{L}^\alpha \mathcal{T}^\alpha \Psi_0 = \max_\pi \langle \pi, R + \gamma P V_0 \rangle + \alpha \mathcal{H}(\pi) \leq \|R + \gamma P V_0\|_\infty + \alpha \log|\mathcal{A}|
$$

and then for pointwise

$$
V_1 - V_0 \leq R_{\max} + 2\|V_0\|_\infty + \alpha \log|\mathcal{A}|.
$$

Combining above and (14), we have

$$V_{k+1} \geq V_0 - \frac{1}{1-\gamma}\left(R_{\max} + 2\left\|V_0\right\|_\infty + \alpha \log|\mathcal{A}|\right) \tag{15}$$

$$\geq -\frac{1-\gamma}{1-\gamma}\left\|V_0\right\|_\infty - \frac{1}{1-\gamma}\left(R_{\max} + 2\left\|V_0\right\|_\infty + \alpha \log|\mathcal{A}|\right) \tag{16}$$

$$\geq -\frac{1}{1-\gamma}\left[3\left\|V_0\right\|_\infty + R_{\max} + \alpha \log|\mathcal{A}|\right]. \tag{17}$$

Now assume that the upper bound of (13) holds up to $k \in \mathbb{N}$. Then we have

$$V_{k+1} = \mathbb{L}^\alpha \mathcal{T}' \Psi_k \leq \mathbb{L}^\alpha \mathcal{T}^\alpha \Psi_k$$
$$= \max \langle \pi, R + \gamma P V_k \rangle + \alpha \mathcal{H}(\pi)$$
$$\leq R_{\max} + \gamma \left\|V_k\right\|_\infty + \alpha \log|\mathcal{A}|$$
$$\leq R_{\max} + \frac{\gamma}{1-\gamma}\left[3\left\|V_0\right\|_\infty + R_{\max} + \alpha \log|\mathcal{A}|\right] + \alpha \log|\mathcal{A}|$$
$$\leq \frac{\gamma}{1-\gamma} 3 \left\|V_0\right\|_\infty + \left(1 + \frac{\gamma}{1-\gamma}\right)\left(R_{\max} + \alpha \log|\mathcal{A}|\right)$$
$$\leq \frac{1}{1-\gamma}\left[3\left\|V_0\right\|_\infty + R_{\max} + \alpha \log|\mathcal{A}|\right]$$

The claim follows since (13) holds for $k = 0$. ∎

**Theorem 5** (Theorem 2 in the main text). *Let $\Psi \in \mathbb{R}^{\mathcal{S} \times \mathcal{A}}$, $V = \mathbb{L}^\alpha \Psi$, $\mathcal{T}^\alpha \Psi = R + \gamma P \mathbb{L}^\alpha \Psi$ and $\mathcal{T}'$ be an operator with the properties that $\mathcal{T}' \Psi \leq \mathcal{T}^\alpha \Psi$ and $\mathcal{T}' \Psi \geq \mathcal{T}^\alpha \Psi - \beta(V - \Psi)$. Consider the sequence $\Psi_{k+1} := \mathcal{T}' \Psi_k$ with $\Psi_0 \in \mathbb{R}^{\mathcal{S} \times \mathcal{A}}$, and let $V_k = \mathbb{L}^\alpha \Psi_k$. Further, with an abuse of notation, we write $V_\tau^* \in \mathbb{R}^{\mathcal{S}}$ as the unique fixed point of the operator $\mathcal{T}^\tau V = \mathbb{L}^\tau (R + \gamma P V)$. Then, the sequence $(V_k)_{k\in\mathbb{N}}$ converges, and the limit $\tilde{V} = \lim_{k\to\infty} V_k$ satisfies $V_\tau^* \leq \tilde{V} \leq V_\alpha^*$. Furthermore, $\limsup_{k\to\infty} \Psi_k \leq Q_\alpha^*$ and $\liminf_{k\to\infty} \Psi_k \geq \frac{1}{1-\beta}\left(\tilde{Q} - \beta\tilde{V}\right)$, where $\tilde{Q} = R + \gamma P \tilde{V}$.*

*Proof.* From (12), we already have the upper bound $\tilde{\Psi} := \limsup_{k\to\infty} \Psi_k \leq Q_\alpha^*$. Now, it holds that

$$\Psi_{k+1} = \mathcal{T}' \Psi_k$$
$$\geq \mathcal{T}^\alpha \Psi_k - \beta(V_k - \Psi_k)$$
$$= R + \gamma P V_k - \beta V_k + \beta \Psi_k. \tag{18}$$

Since $\mathbb{L}^\alpha \Psi = \alpha \log \langle \mathbf{1}, \exp \Psi/\alpha \rangle$ is continuous w.r.t. $\Psi$, Lemma 6 implies that the sequence $(\Psi_k)_{k\in\mathbb{N}}$ is bounded. Now, $V_k$ converges to $\tilde{V}$ by Lemma 5. Furthermore, by Lemma 6 and Lebesgue's dominated convergence theorem, we have

$$\lim_{k\to\infty} P V_k = P \tilde{V}. \tag{19}$$

Taking the $\limsup$ of both sides of (18), we obtain

$$\tilde{\Psi} \geq R + \gamma P \tilde{V} - \beta \tilde{V} + \beta \tilde{\Psi}$$
$$= \tilde{Q} - \beta \tilde{V} + \beta \tilde{\Psi},$$

where $\tilde{Q} = R + \gamma P \tilde{V}$. Thus it holds that

$$\tilde{\Psi} \geq \frac{1}{1-\beta}\left(\tilde{Q} - \beta\tilde{V}\right). \tag{20}$$

In addition, from the fact $\liminf_{k\to\infty} V_k = \tilde{V}$ and taking the $\liminf$ of both sides of (18), which Lemma 6 guarantees to exist again, we also obtain the lower bound of $\liminf_{k\to\infty} \Psi_k$:

$$\liminf_{k\to\infty} \Psi_k \geq \frac{1}{1-\beta}\left(\tilde{Q} - \beta\tilde{V}\right).$$

Applying $\mathbb{L}^\alpha$ to the both sides of (20) and from Lemma 4, it follows that

$$\tilde{V} \geq \mathbb{L}^\tau \tilde{Q} = \mathbb{L}^\tau \left( R + \gamma P \tilde{V} \right) = \mathcal{T}^\tau \tilde{V}.$$

Using the above recursively, we have

$$\tilde{V} \geq \lim_{k \to \infty} (\mathcal{T}^\tau)^k \tilde{V} = V_\tau^*. \tag{21}$$

Now, since $\mathbb{L}^\alpha \Psi$ is continuous w.r.t. $\Psi$ and strictly increasing everywhere, it holds that

$$\limsup_{k \to \infty} V_k = \limsup_{k \to \infty} \mathbb{L}^\alpha \Psi_k = \mathbb{L}^\alpha \limsup_{k \to \infty} \Psi_k \leq \mathbb{L}^\alpha Q_\alpha^* = V_\alpha^*. \tag{22}$$

Combining (21) and (22), we have

$$V_\tau^* \leq \tilde{V} \leq V_\alpha^*.$$

∎

## A.4 PROOF OF PROPOSITION 1

We provide several lemmas that are used to prove Theorem 1.

**Lemma 7.** *Consider the sequence* $\Psi_{k+1} := \mathcal{T}_{\pi_{k+1}}^{fg} \Psi_k$ *produced by the BAL operator* (8) *with* $\Psi_0 \in \mathbb{R}^{\mathcal{S} \times \mathcal{A}}$, *and let* $V_k = \mathbb{L}^\alpha \Psi_k$. *Then the sequence* $(V_k)_{k \in \mathbb{N}}$ *converges, if it holds that*

$$\lambda D_{\mathrm{KL}}(\pi_{k+1} \| \pi_k) - \gamma P^{\pi_{k+1}} \left( \alpha \mathcal{H}(\pi_{k+1}) + \langle \pi_{k+1}, g(A_k) \rangle \right) \geq 0 \tag{23}$$

*for all* $k \in \mathbb{N}$.

*Proof.* We follow similar steps as in the proof of Lemma 5. First, since $\mathcal{T}_{\pi_{k+1}}^{fg} \Psi_k \leq \mathcal{T}^\alpha \Psi_k$ we have $\limsup_{k \to \infty} \Psi_k =: \tilde{\Psi} \leq Q_\alpha^*$. Let $\tilde{V} := \limsup_{k \to \infty} V_k$. Now, it holds that

$$\begin{aligned}
V_{k+1} = \mathbb{L}^\alpha \Psi_{k+1} &= \langle \pi_{k+2}, \Psi_{k+1} \rangle + \alpha \mathcal{H}(\pi_{k+2}) \\
&\geq \langle \pi_{k+1}, \Psi_{k+1} \rangle + \alpha \mathcal{H}(\pi_{k+1}) \\
&= \left\langle \pi_{k+1}, \mathcal{T}_{\pi_{k+1}}^{fg} \Psi_k \right\rangle + \alpha \mathcal{H}(\pi_{k+1}) \\
&= \langle \pi_{k+1}, \mathcal{T}_{\pi_{k+1}} \Psi_k - \gamma P \langle \pi_{k+1}, g(A_k) \rangle + \beta f(A_k) \rangle + \alpha \mathcal{H}(\pi_{k+1}) \\
&\overset{(a)}{\geq} \langle \pi_{k+1}, \mathcal{T}_{\pi_{k+1}} \Psi_k - \gamma P \langle \pi_{k+1}, g(A_k) \rangle + \beta A_k \rangle + \alpha \mathcal{H}(\pi_{k+1}) \\
&\overset{(b)}{=} \langle \pi_{k+1}, \mathcal{T}_{\pi_{k+1}} \Psi_k \rangle + \tau \mathcal{H}(\pi_{k+1}) - \gamma \langle \pi_{k+1}, P \langle \pi_{k+1}, g(A_k) \rangle \rangle \\
&\overset{(c)}{=} \langle \pi_{k+1}, R + \gamma P (V_k - \alpha \mathcal{H}(\pi_{k+1})) \rangle + \tau \mathcal{H}(\pi_{k+1}) - \gamma P^{\pi_{k+1}} \langle \pi_{k+1}, g(A_k) \rangle \\
&\overset{(d)}{=} \langle \pi_{k+1}, Q_k + \gamma P(V_k - V_{k-1}) \rangle + \tau \mathcal{H}(\pi_{k+1}) - \gamma P^{\pi_{k+1}} \left( \alpha \mathcal{H}(\pi_{k+1}) + \langle \pi_{k+1}, g(A_k) \rangle \right) \\
&\overset{(e)}{=} V_k + \gamma P^{\pi_{k+1}}(V_k - V_{k-1}) + \lambda D_{\mathrm{KL}}(\pi_{k+1} \| \pi_k) - \gamma P^{\pi_{k+1}} \left( \alpha \mathcal{H}(\pi_{k+1}) + \langle \pi_{k+1}, g(A_k) \rangle \right),
\end{aligned}$$

where (a) follows from the non-negativity of the advantage $A_k$ and $x - f(x) \leq 0$, where (b) follows from $\langle \pi_{k+1}, A_k \rangle = \langle \pi_{k+1}, \alpha \log \pi_{k+1} \rangle = -\alpha \mathcal{H}(\pi_{k+1})$ and $(1 - \beta)\alpha = \tau$, (c) follows from $V_k = \mathbb{L}^\alpha \Psi_k = \langle \pi_{k+1}, \Psi_k \rangle + \alpha \mathcal{H}(\pi_{k+1})$, (d) follows from $\mathcal{T}^\alpha \Psi_k = R + \gamma P \mathbb{L}^\alpha \Psi_k = R + \gamma P V_k = Q_{k+1}$, and (e) follows from $V_k = \mathbb{L}^\alpha \Psi_k = \langle \pi_{k+1}, Q_k \rangle + \tau \mathcal{H}(\pi_{k+1}) - \lambda D_{\mathrm{KL}}(\pi_{k+1} \| \pi_k)$. Thus, if it holds that

$$\lambda D_{\mathrm{KL}}(\pi_{k+1} \| \pi_k) - \gamma P^{\pi_{k+1}} \left( \alpha \mathcal{H}(\pi_{k+1}) + \langle \pi_{k+1}, g(A_k) \rangle \right) \geq 0$$

for all $k$, we have

$$V_{k+1} - V_k \geq \gamma P^{\pi_{k+1}}(V_k - V_{k-1}).$$

Therefore, by following the steps equivalent to the proof of Lemma 5, we have that $\liminf_{k \to \infty} V_k = \tilde{V}$ and $V_k$ converges. ∎

**Lemma 8.** *Let the conditions of Lemma 7 holds. Then for all $k \in \mathbb{N}$,*

$$|V_k| \leq \frac{1}{1-\gamma}\Big[3\,\|V_0\|_\infty + R_{\max} + \alpha \log|\mathcal{A}|\Big]. \tag{24}$$

*Proof.* Since the proof of Lemma 6 relies on two inequalities $\mathcal{T}'\Psi \leq \mathcal{T}^\alpha\Psi$ and $V_{k+1} - V_k \geq \gamma P^{\pi_{k+1}}(V_k - V_{k-1})$, the claim follows from the identical steps. ∎

We are ready to prove Proposition 1.

**Proposition 2** (Proposition 1 in the main text). *Consider the sequence $\Psi_{k+1} := \mathcal{T}^{fg}_{\pi_{k+1}}\Psi_k$ produced by the BAL operator (8) with $\Psi_0 \in \mathbb{R}^{\mathcal{S}\times\mathcal{A}}$, and let $V_k = \mathbb{L}^\alpha \Psi_k$. Assume that for all $k \in \mathbb{N}$ it holds that*

$$\lambda D_{\mathrm{KL}}(\pi_{k+1}\|\pi_k) - \gamma P^{\pi_{k+1}}\left(\alpha\mathcal{H}(\pi_{k+1}) + \langle\pi_{k+1}, g(A_k)\rangle\right) \geq 0. \tag{25}$$

*Then, the sequence $(V_k)_{k\in\mathbb{N}}$ converges, and the limit $\tilde{V} = \lim_{k\to\infty} V_k$ satisfies $V^*_\tau - \frac{\gamma}{1-\gamma}\frac{\alpha}{1-\beta}\log|\mathcal{A}| \leq \tilde{V} \leq V^*_\alpha$. Furthermore, $\limsup_{k\to\infty}\Psi_k \leq Q^*_\alpha$ and $\liminf_{k\to\infty}\Psi_k \geq \frac{1}{1-\beta}\left(\tilde{Q} - \beta\tilde{V} - \gamma\alpha\log|\mathcal{A}|\right)$, where $\tilde{Q} = R + \gamma P\tilde{V}$.*

*Proof.* We already have the upper bound $\tilde{\Psi} := \limsup_{k\to\infty}\Psi_k \leq Q^*_\alpha$. It holds that

$$\begin{aligned}
\Psi_{k+1} &= \mathcal{T}^{fg}_{\pi_{k+1}}\Psi_k \\
&= \mathcal{T}_{\pi_{k+1}}\Psi_k - \gamma P\langle\pi_{k+1}, g(A_k)\rangle + \beta f(A_k) \\
&\overset{(a)}{\geq} \mathcal{T}_{\pi_{k+1}}\Psi_k + \beta(V_k - \Psi_k) \\
&= R + \gamma PV_k - \beta V_k + \beta\Psi_k - \gamma\alpha P\mathcal{H}(\pi_{k+1}) \\
&\geq R + \gamma PV_k - \beta V_k + \beta\Psi_k - \gamma\alpha\log|\mathcal{A}|, \tag{26}
\end{aligned}$$

where (a) follows from the non-positivity of the soft advantage and the property of $f$ and $g$. Since $\mathbb{L}^\alpha\Psi = \alpha\log\langle\mathbf{1}, \exp\Psi/\alpha\rangle$ is continuous w.r.t. $\Psi$, Lemma 8 implies that the sequence $(\Psi_k)_{k\in\mathbb{N}}$ is bounded. Now, $V_k$ converges to $\tilde{V}$ by Lemma 7. Furthermore, by Lemma 8 and Lebesgue's dominated convergence theorem, we have $\lim_{k\to\infty} PV_k = P\tilde{V}$. Let $\bar{\Psi} := \liminf_{k\to\infty}\Psi_k$. Taking the $\liminf$ of both sides of (26), we obtain

$$\begin{aligned}
\bar{\Psi} &\geq R + \gamma P\tilde{V} - \beta\tilde{V} + \beta\bar{\Psi} - \gamma\alpha\log|\mathcal{A}| \\
&= \tilde{Q} - \beta\tilde{V} + \beta\bar{\Psi} - \gamma\alpha\log|\mathcal{A}|,
\end{aligned}$$

where $\tilde{Q} = R + \gamma P\tilde{V}$. Thus it holds that

$$\bar{\Psi} \geq \frac{1}{1-\beta}\left(\tilde{Q} - \beta\tilde{V} - \gamma\alpha\log|\mathcal{A}|\right).$$

Now, applying $\mathbb{L}^\alpha$ to the both sides of the above and follwoing the argument to derive (21), we have

$$\tilde{V} \geq \mathbb{L}^\tau\tilde{Q} - \frac{\gamma\alpha}{1-\beta}\log|\mathcal{A}| = \mathcal{T}^\tau\tilde{V} - \frac{\gamma\alpha}{1-\beta}\log|\mathcal{A}|,$$

where we used the fact that $\mathbb{L}^\alpha(Q+c) = \mathbb{L}^\alpha(Q)+c$ for a constant $c$. Therefore, using this expression recursively we obtain

$$\tilde{V} \geq V^*_\tau - \frac{\gamma}{1-\gamma}\frac{\alpha}{1-\beta}\log|\mathcal{A}|.$$

Furthermore, since $\tilde{\Psi} = \limsup_{k\to\infty}\Psi_k \leq Q^*_\alpha$ we have $\limsup_{k\to\infty} V_k \leq V^*_\alpha$ again. ∎

## A.5 PROOF OF THEOREM 3

**Theorem 6** (Theorem 3 in the main text). *Let $\{\pi_k\}_k$ be a sequence of the policies obtained by BAL. Defining $\Delta_k^{fg} = \langle \pi^*, \beta\left(A_\tau^* - f(A_{k-1})\right) - \gamma P \langle \pi_k, A_{k-1} - g(A_{k-1})\rangle\rangle$, it holds that:*

$$\left\| V_\tau^* - V_\tau^{\pi_{K+1}} \right\|_\infty \le \frac{2\gamma}{1-\gamma}\left[ 2\gamma^{K-1} V_{\max}^\tau + \sum_{k=1}^{K-1} \gamma^{K-k-1} \left\| \Delta_k^{fg} \right\|_\infty \right]. \tag{27}$$

*Proof.* For the policy $\pi_{k+1} = \mathcal{G}^{0,\alpha}(\Psi_k)$, the operator $\mathcal{T}_{\pi_{k+1}}^{0,\tau}$ is a contraction map. Let $V_\tau^{\pi_{K+1}}$ denote the fixed point of $\mathcal{T}_{\pi_{K+1}}^{0,\tau}$, that is, $V_\tau^{\pi_{K+1}} = \mathcal{T}_{\pi_{K+1}}^{0,\tau} V_\tau^{\pi_{K+1}}$. Observing that $\pi_{k+1} = \mathcal{G}_{\pi_k}^{\lambda,\tau}(Q_k) = \mathcal{G}_{\pi_k}^{\lambda,\tau}(R + \gamma P V_{k-1})$, we have for $K \ge 1$,

$$V_\tau^* - V_\tau^{\pi_{K+1}} = \mathcal{T}_{\pi^*}^{0,\tau} V_\tau^* - \mathcal{T}_{\pi^*}^{0,\tau} V_{K-1} + \mathcal{T}_{\pi^*}^{0,\tau} V_{K-1} - \mathcal{T}^\tau V_{K-1} + \mathcal{T}^\tau V_{K-1} - \mathcal{T}_{\pi_{K+1}}^{0,\tau} V_\tau^{\pi_{K+1}}$$

$$\overset{(a)}{\le} \gamma P^{\pi^*}(V_\tau^* - V_{K-1}) + \gamma P^{\pi_{K+1}}(V_{K-1} - V_\tau^{\pi_{K+1}})$$

$$= \gamma P^{\pi^*}(V_\tau^* - V_{K-1}) + \gamma P^{\pi_{K+1}}(V_{K-1} - V_\tau^* + V_\tau^* - V_\tau^{\pi_{K+1}})$$

$$= (I - \gamma P^{\pi_{K+1}})^{-1}\left(\gamma P^{\pi^*} - \gamma P^{\pi_{K+1}}\right)(V_\tau^* - V_{K-1}), \tag{28}$$

where (a) follows from $\mathcal{T}_{\pi^*}^{0,\tau} V_{K-1} \le \mathcal{T}^\tau V_{K-1} = \mathcal{T}_{\pi_{K+1}}^{0,\tau} V_{K-1}$ and the definition of $\mathcal{T}_\pi^{0,\tau}$.

We proceed to bound the term $V_\tau^* - V_{K-1}$:

$$V_\tau^* - V_{K-1} = \mathcal{T}_{\pi^*}^{0,\tau} V_\tau^* - \mathcal{T}_{\pi^*}^{0,\tau} V_{K-2} + \mathcal{T}_{\pi^*}^{0,\tau} V_{K-2} - \mathbb{L}^\alpha \Psi_{K-1}$$

$$= \gamma P^{\pi^*}(V_\tau^* - V_{K-2}) + \Delta_{K-1},$$

where $\Delta_{K-1} = \mathcal{T}_{\pi^*}^{0,\tau} V_{K-2} - \mathbb{L}^\alpha \Psi_{K-1}$. Observing that

$$\mathbb{L}^\alpha \Psi_{K-1} = \langle \pi_K, \Psi_{K-1}\rangle + \alpha \mathcal{H}(\pi_K)$$

$$= \max_\pi \langle \pi, \Psi_{K-1}\rangle + \alpha \mathcal{H}(\pi)$$

$$\ge \langle \pi^*, \Psi_{K-1}\rangle + \alpha \mathcal{H}(\pi^*)$$

$$= \langle \pi^*, R + \beta f(A_{K-2}) + \gamma P \langle \pi_{K-1}, \Psi_{K-2} - g(A_{K-2})\rangle\rangle + (\tau + \beta\alpha)\mathcal{H}(\pi^*),$$

we have

$$\Delta_{K-1} = \langle \pi^*, R + \gamma P V_{K-2}\rangle + \tau \mathcal{H}(\pi^*) - \mathbb{L}^\alpha \Psi_{K-1}$$

$$\le \langle \pi^*, \gamma P V_{K-2}\rangle - \langle \pi^*, \beta f(A_{K-2}) + \gamma P \langle \pi_{k-1}, \Psi_{K-2} - g(A_{K-2})\rangle\rangle - \beta\alpha\mathcal{H}(\pi^*)$$

$$= \langle \pi^*, \beta\left(A_\tau^* - f(A_{K-2})\right) - \gamma P \langle \pi_{K-1}, A_{K-2} - g(A_{K-2})\rangle\rangle$$

$$=: \Delta_{K-1}^{fg}.$$

Thus, it follows that

$$V_\tau^* - V_{K-1} \le \gamma P^{\pi^*}(V_\tau^* - V_{K-2}) + \Delta_{K-1}^{fg}$$

$$\le \left(\gamma P^{\pi^*}\right)^{K-1}(V_\tau^* - V_0) + \sum_{k=1}^{K-1}\left(\gamma P^{\pi^*}\right)^{K-k-1} \Delta_k^{fg}.$$

Plugging the above into (28) and taking $\|\cdot\|_\infty$ on both sides, we obtain

$$\left\| V_\tau^* - V_\tau^{\pi_{K+1}} \right\|_\infty \le \frac{2\gamma}{1-\gamma}\left[ 2\gamma^{K-1} V_{\max}^\tau + \sum_{k=1}^{K-1} \gamma^{K-k-1} \left\| \Delta_k^{fg} \right\|_\infty \right]. \tag{29}$$

∎

# B  ADDITIONAL EXPERIMENTAL DETAILS.

## B.1  BAL ON GRID WORLD.

Figure 10 shows the grid world environment used in Section 5.1. The reward is $r = 1$ at the top-right and botom left corners, $r = 2$ at the bottom-right corner and $r = 0$ otherwise. The action space is $\mathcal{A} = \{\text{North}, \text{South}, \text{West}, \text{East}\}$. An attempted action fails with probability 0.1 and random action is performed uniformly. We set $\gamma = 0.99$. We chose $\alpha = 0.02$ and $\beta = 0.99$, thus $\tau = (1 - \beta)\alpha = 0.0002$ and $\lambda = \beta\alpha = 0.0198$. Since the transition kernel $P$ and the reward function $R$ are directly available for this environment, we can perform the model-based M-VI (2) and BAL (9) schemes. We performed 100 independent runs with random initialization of $\Psi$ by $\Psi_0(s, a) \sim \text{Unif}(-V_{\max}^\tau, V_{\max}^\tau)$. Figure 4 compares the normalized value of the suboptimality $\|V^{\pi_k} - V_\tau^*\|_\infty$, where we computed $V_\tau^*$ by the recursion $V_{k+1} = \mathcal{T}^\tau V_k = \mathbb{L}^\tau(R + \gamma P V_k)$ with $V_0(s) = 0$ for all state $s \in \mathcal{S}$.

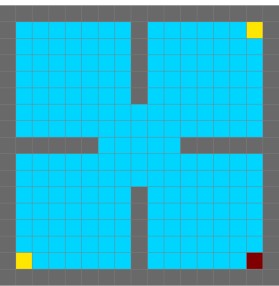

Figure 10: Grid world environment for model-based experiment.

## B.2  MDAC ON MUJOCO.

We used PyTorch[2] and Gymnasium[3] for all the experiments. We used rliable[4] to calculate the IQM scores. MDAC is implemented based on SAC agent from CleanRL[5]. Each trial of MDAC run was performed by a single NVIDIA V100 with 8 CPUs and took approximately 8 hours for 3M environment steps. For the baselines, we used SAC agent from CleanRL with default parameters from the original paper. We used author's implementation[6] for TD3 with default parameters.

Table 1 summarizes the hyperparameter values for MDAC, which are equivalent to the values for SAC except the additional $\beta$.

**Per-environment results.** Here, we provide per-environment results for ablation studies. Figure 12, 13, 14 and 15 show the per-environment results for Figure 5, 6, 8 and. 9, respectively.

**Quantities in TD target under clipping.** Figure 16 shows the the quantities in TD target for $f = g = \text{clip}(x, -1, 1)$ and $f = g = \text{clip}(x/10, -1, 1)$.

---

[2]https://github.com/pytorch/pytorch
[3]https://github.com/Farama-Foundation/Gymnasium
[4]https://github.com/google-research/rliable
[5]https://github.com/vwxyzjn/cleanrl
[6]https://github.com/sfujim/TD3

Table 1: MDAC Hyperparameters

| Parameter | Value |
|---|---|
| optimizer | Adam (Kingma & Ba, 2015) |
| learning rate | $3 \cdot 10^{-4}$ |
| discount factor $\gamma$ | 0.99 |
| replay buffer size | $10^6$ |
| number of hidden layers (all networks) | 2 |
| number of hidden units per layer | 256 |
| number of samples per minibatch | 256 |
| nonlinearity | ReLU |
| target smoothing coefficient by polyack averaging ($\kappa$) | 0.005 |
| target update interval | 1 |
| gradient steps per environmental step | 1 |
| reparameterized KL coefficient $\beta$ | $1 - (1 - \gamma)^2$ |
| entropy target $\bar{\mathcal{H}}$ to optimize $\tau = (1 - \beta)\alpha$ | $-\dim(\mathcal{A})$ |

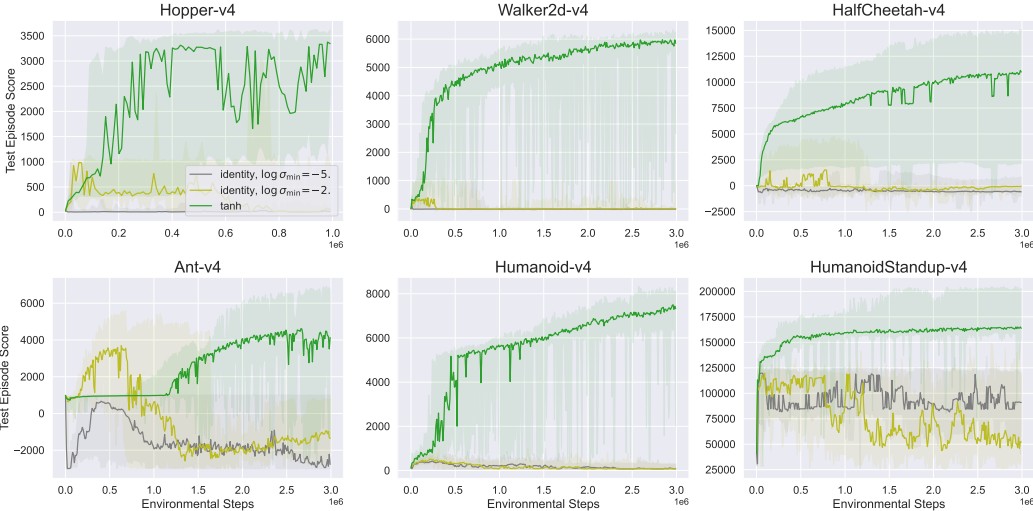

Figure 11: Per-environment performances for Figure 1. The median scores of 10 independent runs are reported. The shaded region corresponds to the minimum and maximum scores over the 10 runs.

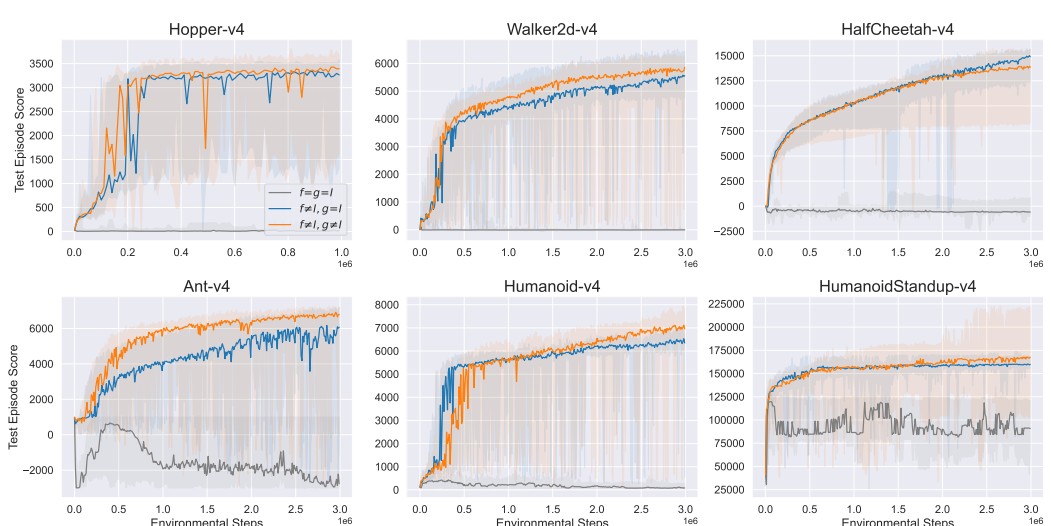

Figure 12: Per-environment performances for Figure 5. The median scores of 10 independent runs are reported. The shaded region corresponds to the minimum and maximum scores over the 10 runs.

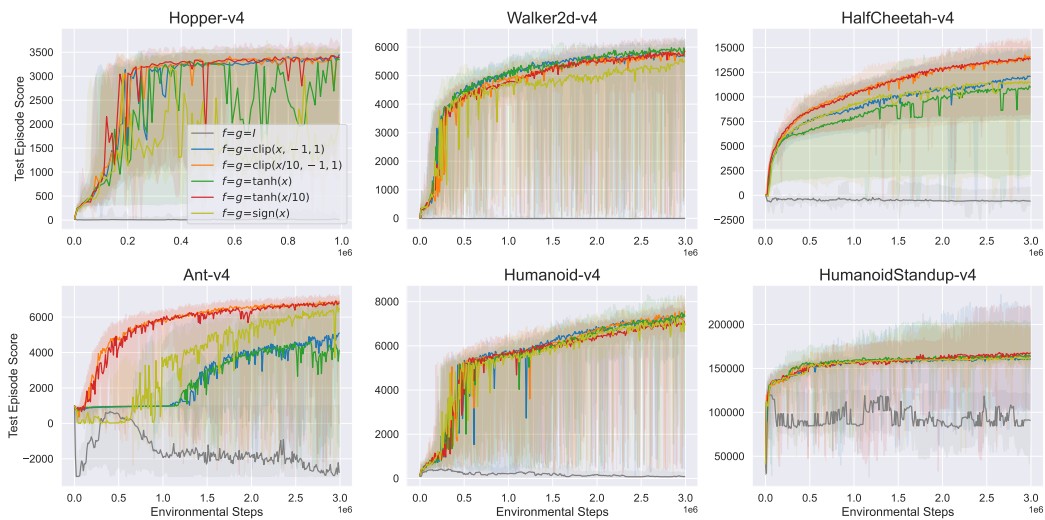

Figure 13: Per-environment performances for Figure 6. The median scores of 10 independent runs are reported. The shaded region corresponds to the minimum and maximum scores over the 10 runs.

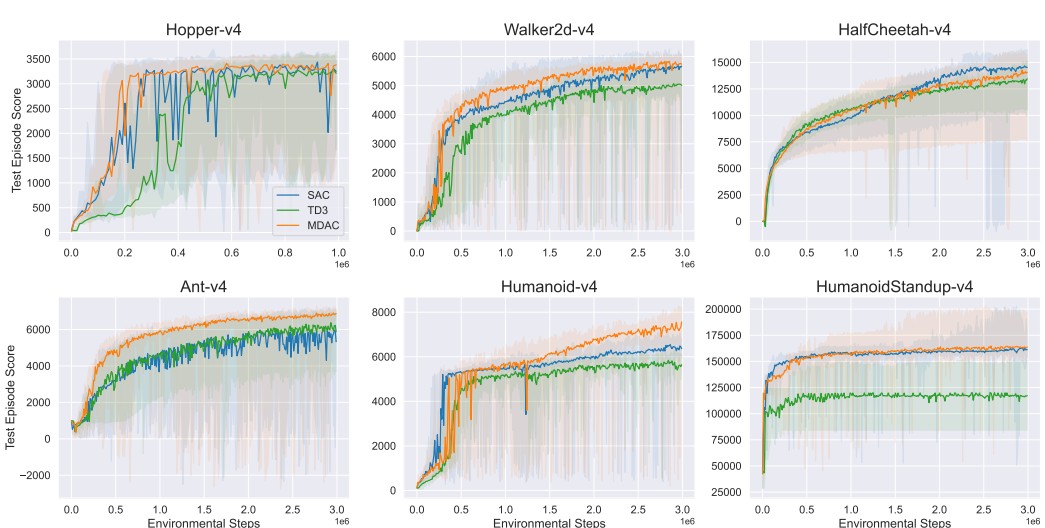

Figure 14: Per-environment performances. The median scores of 10 independent runs are reported. The shaded region corresponds to the minimum and maximum scores over the 10 runs.

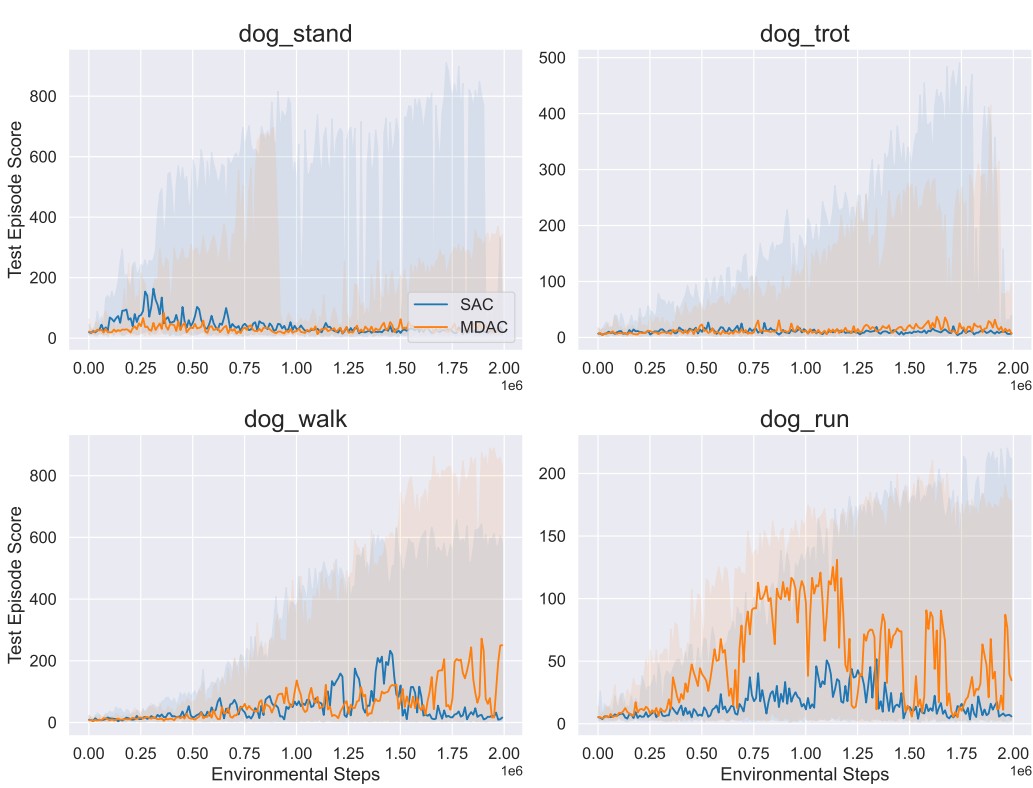

Figure 15: Per-environment performances in `dog` domain from DeepMind Control Suite. The median scores of 10 independent runs are reported. The shaded region corresponds to the minimum and maximum scores over the 10 runs.

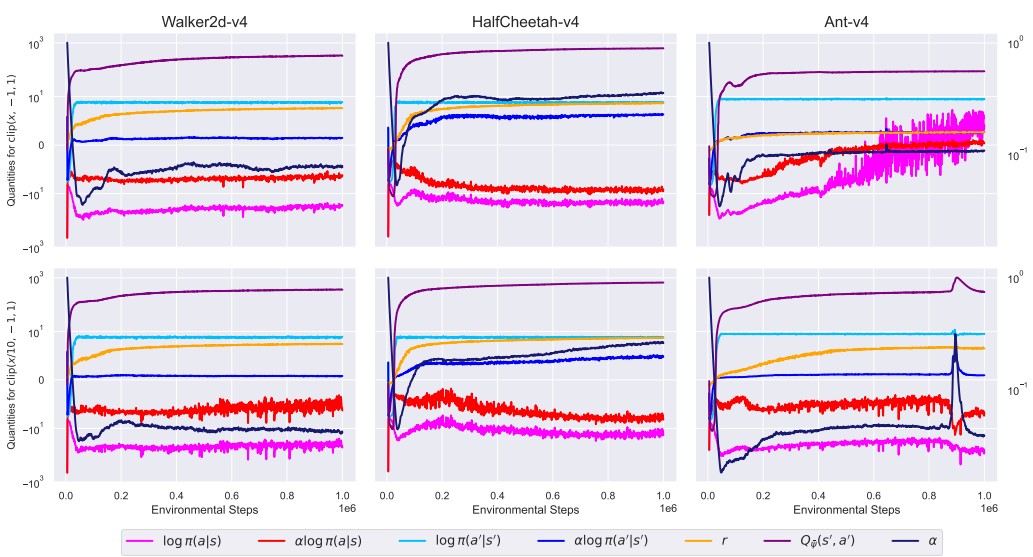

Figure 16: Scale comparison of the quantities in TD target. Top row: clip$(x, -1, 1)$, Bpttom row: clip$(x/10, -1, 1)$, Left column: Walker2d-v4, Middle column: HalfCheetah-v4. Right column: Ant-v4.

