# OpenReview forum: "Mirror Descent Actor Critic via Bounded Advantage Learning"
_ICLR.cc/2025/Conference — Submitted to ICLR 2025_

### Official Review · Reviewer_NWya · 2024-10-21

**Soundness:** 3
**Presentation:** 2
**Contribution:** 3
**Rating:** 6
**Confidence:** 2

**Summary:**

This paper proposes Mirror Descent Actor-Critic (MDAC), which extends Mirror Descent Value Iteration (MDVI) into continuous action domains. The authors showed that naively implementing MDAC in practice can lead to instability, and found that this is caused by the magnitude of the log policy terms being much larger than the magnitude of the rewards. As a fix, the authors proposed to bound the effect of the log policy terms by transforming them through a bounded function, and showed that this can recover the agent’s performance. The authors then demonstrated that this ad-hoc fix can be seen as an instantiation of advantage learning, Lastly, the authors demonstrated that the proposed approach is competitive with previous methods like SAC and TD3.

**Strengths:**

1. The paper extends MDVI style regularization, which was found to be useful in the discrete action domains, to continuous action domains. This can potentially enable more robust design of algorithms, and is certainly a valuable contribution to the community.
2. The proposed solution (i.e., bounding the effect of the log policy terms) is backed by both theory and empirical evidence (mujoco experiments). In addition, the theory can also explain certain design decisions made in previous works (e.g., the log policy clipping used by munchausen DQN).

**Weaknesses:**

My main concern is that it is not clear to me whether the proposed fix (bounding the log policy terms through transformations) is more effective than simply constraining the policy (e.g., lower bounding the standard deviation of the Gaussian policy) to avoid this problem in the first place.

Since SAC is a special case of MDAC, I don’t understand why SAC is not suffering from the same problem of exploding log policies. After digging through the code provided by the authors, I think this is partially due to the authors using a much lower bound on the log_std parametrization. For example, in the CleanRL SAC implementation [1], log_std_min is set at -5, while the authors set this parameter at -20. As another example, the official implementation explicitly lower bounds the standard deviation at 1e-5 [2], which is much larger than the value used by the authors. This might explain why, in Figure 3, the log policies are showing extreme values. To test this, I would suggest the authors to reproduce Figure 3 for different values of log_std_min to see its impact.

[1] https://github.com/vwxyzjn/cleanrl/blob/38c313f8326b5049fe941a873e798485bccf18e5/cleanrl/sac_continuous_action.py#L97

[2] https://github.com/rail-berkeley/softlearning/blob/13cf187cc93d90f7c217ea2845067491c3c65464/softlearning/policies/gaussian_policy.py#L276

**Questions:**

1. On line 349, it appears that using functions with smaller $c_f$ can improve convergence speed, then why not simply use $f\equiv 0$? What would be the trade-off here?
2. In contrast to previous methods, the sub-optimality of BAL (Equation 10) does not seem to explicitly depend on the value estimation error $\epsilon_k$. Can the authors elaborate a bit on this?

---

> ### Author Response · Authors · 2024-11-13
> **Thank you for your comment.**
>
> Thank you for reviewing our paper. We sincerely appreciate your comments.
>
> ### On Weaknesses
>
> Thank you for pointing out the important issue. Since the reviewer's concern is very important in continuous action settings, we will run additional experiments in this discussion period, and try to reproduce Figure 3 for different values of `log_std_min` to see its impact.
>
> The followings are our current answers to your concern, that we can address before performing the additional experiments.
> - First, we would like to emphasize that, the bounding is effective even in the model-based tabular setting (Figure 4), where the "constraining the policy" does not apply.
> - We remark that, our choice `log_std_min=-20` is not odd. Indeed, OpenAI's SpinningUp implementation chooses `-20` as well [1].
> - Figure 3 and 8 indicate that, the problem of naive MDAC comes largely from the naive Munchausen bonus term, which is evaluated for the off-policy $(s,a)$ pairs in the replay buffer. On the other hand, the entropy bonus term is evaluated for the successor $(s',a')$ pairs, where $a'$ is sampled from the current actor. Thanks to this "on-policyness" of $a'$, the explosion of the entropy bonus term is not as severe as the Munchausen bonus term. We believe that this is why SAC does not suffer from the explosion of log policy as seriously as MDAC.
> - We also would like to clarify that, to acquire the learning results for SAC, we used CleanRL's SAC without modifying; we used `log_std_min=-5`.
>
>
> ### Answers to Questions
> 1. If we choose $c_f=0$, BAL reduces to an entropy-only-regularized scheme without KL regularization. Since our main interest is in the improvement of both KL-entropy regularized RL, we excluded the choice $c_f=0$. Indeed, our experimental result suggests that MDAC performs better than SAC, an instantiation of entropy-only-regularized scheme.
> 2. Thank you for seeking for clarifying this point. Our initial submission lacks to explicitly explain this. Our error terms do not have the approximation/estimation error $\epsilon_k$, because we simply omitted it in our analysis. To be precise, at L.1005 in Appendix A.4, we used $
>   \Psi_{K-1}
>   = R + \beta {f}(A_{K-2}) + \gamma P \left<\pi_{K-1}, \Psi_{K-2} - {g}(A_{K-2})\right>
> $
>  excluding $\epsilon_k$.
> If we include $\epsilon_k$ to our analysis, the definition of $\Delta_{k}^{fg}$ will be replaced by
> $\Delta_{k}^{fg} = \left<\pi^\ast,
>   \beta \left(A_{\tau}^\ast - {f}(A_{k-1})\right) - \gamma P \left<
>     \pi_{k}, A_{k-1} - {g}(A_{k-1})\right> + \epsilon_k \right>
> $.
> By using this expression, we can discuss the effect of $\epsilon_k$ to BAL as well. However, since our main interest in Section 4 is not in the effect of the approximation/estimation error, but in the effect of the optimal policy misspecification inherent to the soft-gap-increasing nature of M-VI and BAL in model-based tabular setting without any approximation, we omitted $\epsilon_{k}$. We will update the manuscript to explain this point.
>
> [1] https://github.com/openai/spinningup/blob/038665d62d569055401d91856abb287263096178/spinup/algos/pytorch/sac/core.py#L27

---

> ### Author Response · Authors · 2024-11-15
> **Additional Reply**
>
> After reviewing our code, we found that all the reported experimental results for MDAC in our manuscript was computed under `log_std_min=-5` as well. We are embarrassed that our policy class `SquashedGaussianPolicy` was not using the externally set value of  `log_std_min`. Please kindly check __L174-L179__ of `mdac_bal/algos/policy.py` of our submitted code.
> Considering this, we can conclude that, the answer to your question, "why SAC is not suffering from the same problem of exploding log policies" compared to MDAC, is the aforementioned "on-policyness" of $a'$ in the entropy bonus term. Thank you again for seeking this.

---

> ### Author Response · Authors · 2024-11-19
> **Update regarding your concerns**
>
> We updated the manuscript. Please kindly check the updated version. The followings are modifications regarding your concerns.
> - We run additional experiments with `log_std_min=-2` and reported the results in __Figure 1__ and __2__. __L168-L183__ explains the learning results and discusses the difference between the Munchasen bonus term and the entropy bonus term. We found that, though the log-policies behave better with `log_std_min=-2`, they still hinder the optimization of $Q_\psi$, $\pi_\theta$ and $\alpha$. The bonus terms $\alpha\log\pi$ explode gradually, which leads to failure again.
> - In __L348-L354__, we explicitly stated that we omitted $\epsilon_{k}$ in our analysis and focus on the inherent error of M-VI and BAL schemes.

---

> > ### Comment · Reviewer_NWya · 2024-11-21
> >
> > Thank you for the clarifications, I think the explanation for why SAC is more stable is reasonable. However, I don't fully understand why, in Figure 2, $\log{\sigma}=-2$  seems more unstable than $\log{\sigma}=-5$. It appears that the main problem is not the log policy terms but rather the $\alpha$ parameter. Can the authors comment on this?

---

> ### Author Response · Authors · 2024-11-21
> **Thank you for your feedback.**
>
> Thank you for your feedback, and thank you for seeking clarification.
> - `log_std_min=-5` is indeed more unstable than `log_std_min=-2`; please kindly check the x-axis of Figure 2, where the left column focuses more earlier stage than the middle and right columns. For `log_std_min=-5`, $\log\pi(a|s)$ explodes to $-10^7$ in $15000$ steps, which is much faster than `log_std_min=-2`, and then $\alpha$ gradually glows after the explosion of $\log\pi(a|s)$. Thus, the pure log policy without $\alpha$ is still harmful for the setting `log_std_min=-5`.
> - For `log_std_min=-2`, we agree that the behavior of $\alpha$ is also problematic. However, since we apply $f,g$ to $\alpha\log\pi$, the bounding can prevent the large value of $\alpha$ from ruining $Q_\psi$.

---

> > ### Comment · Reviewer_NWya · 2024-11-21
> >
> > Thanks again for the clarifications. My concern with the new experiment is that it seems to suggest that the root of the problem is not only the log policy terms, but also the alpha parameter and how it's optimized (Eq5), since the log policy terms for $\log\sigma=-2$ are now on a similar scale compared to the bounded approach before 1e5 steps (for Walker2d), and things started to destabilize only after $\alpha$ started increasing.
> > I'd also like to emphasize that this by no means downplays the approach proposed by the authors, which, I think, is a robust solution to this problem; however, this might require some changes to the narrative of the paper.

---

> ### Author Response · Authors · 2024-11-22
> **Thank you for your feedbacks**
>
> We thank the reviewer again for the kind and constructive feedbacks. We agree that $\alpha$ is also a root cause of naive unstable behavior. We will update the paper so that it better highlight this aspect.
>
> Though we do not have strong evidence to support, the followings are conjecture on the behavior of $\alpha$.
> - The followings are the experimental results acquired so far:
>   - By construction, the direct outcome by bounding is the stabilization of $Q_\psi$ only.
>   - However, as we observe from Figure 2, the bounding stabilizes not only $Q_\psi$ but also $\alpha$.
>   - Thus, the stable behavior of $Q_\psi$ seems the key for stable $\alpha$.
> - On the properties of loss functions:
>   - The $\alpha\log\pi$ is forced to have similar value to $Q_\psi$ through the loss (4).
>   -  On the other hand, though the the loss (5) does not have the explicit notion of the quantities related to reward or value,
> 	from the identity in $\alpha\log\pi=A$ in the tabular case, we can think of the loss (5) as a kind of a constraint to the soft advantage, which requires $\alpha$ to have the value, with which the (scaled) soft advantage matches $\bar{\mathcal{H}}=-{\rm dim}\mathcal{A}$, __regardless__ the scale of reward and values.
> - Give the observations above, we conjecture that, the cause of exploding $\alpha$
> is this target mismatch between (4) and (5); $\log\pi$ tries to stay close to $\bar{\mathcal{H}}$, and $\alpha$ grows large to compensate the scale mismatch with $Q_\psi$.
> By bounding, since $Q_\psi$ behaves stably, $\alpha$ need not to be aggressive for this compensation.

---

> ### Author Response · Authors · 2024-11-27
> **Refined conjecture on the behavior of $\alpha$**
>
> Thank you again for seeking the clarification. We noticed that the phenomenon on $\alpha$ is much simpler.
> - A naive update of $\alpha$ using the loss (5) and SGD is expressed as
> $\alpha \leftarrow \alpha + \frac{\rho (1-\beta)}{N} \sum_{n=1}^{N}  \left( \log \pi_{\theta}(a_n|s_n) - {\rm dim}(\mathcal{A})\right)$, where $\rho>0$ is a step-size, $N$ is a mini-batch size and
> $a_n\sim\pi_{\theta}(\cdot|s_n)$.
> - The above expression indicates that, if the average of $\log\pi_{\theta}(a|s)$ over sampled mini-batches are bigger than ${\rm dim}(\mathcal{A})$, $\alpha$ keeps growing.
> This could be the case when (i) $Q_\psi$ is unstable and (ii) $\log\pi$ is ruined through the loss (4). We will update the manuscript to argue this phenomenon in Section 3.

---

> > ### Comment · Reviewer_NWya · 2024-11-28
> >
> > Thank you for the detailed analysis, I think this is helpful for understanding the cause of the instability.
> > However, as I'm not an expert in the theories of MDVI, I will keep my score/confidence but lean favorably towards acceptance due to the strong empirical evidence provided.

---

> ### Author Response · Authors · 2024-11-29
> **Thank you for your feedbacks**
>
> Thank you for keeping the positive score, and thank you again for your constructive feedbacks. We appreciate that the manuscript has been improved thanks to your comments. Section 3 better motivates to introduce the bounding.

---

### Official Review · Reviewer_tZxa · 2024-10-29

**Soundness:** 3
**Presentation:** 3
**Contribution:** 3
**Rating:** 6
**Confidence:** 3

**Summary:**

This paper introduces Mirror Descent Actor Critic (MDAC), a novel approach to enhance reinforcement learning in continuous action spaces. By bounding the log-density values in the critic's loss function, MDAC significantly boosts performance over traditional entropy-only methods. The authors show that this approach reduces policy error, connecting MDAC to Advantage Learning and providing theoretical support for improved stability in continuous domains.

**Strengths:**

**Good motivation**
- Improvement of the performance of Mirror Descent RL on continuous action tasks.

**Theoretical and Empirical Rigour**
- The progression from MDVI to MDAC is well-motivated, and the authors thoughtfully discuss the relation to SAC’s temperature tuning. I found the performance gap between non-bounded (identity) and bounded (tanh) log-policy to be surprisingly substantial.

**Weaknesses:**

**Section 4**
- It seems that bounding the log-policy was meant to yield a tighter bound in Theorem 3, connecting it to Advantage Learning (AL). Could you clarify if BAL was introduced specifically for this purpose?
- L224: I didn’t see definitions for regularized MDP and soft state value function—could you include these?
- L227: Why does V(s)=max_{⁡a∈A} Q(s,a) hold when α=0?
- L239 (Eq. 9): Could you provide the derivation for this Bellman operator?
- L245: Could you explain how the gap-increasing Bellman operator reduces suboptimal action values?
- L247: I’m unfamiliar with the literature on successor state functions—could you clarify the meaning of successor state-action pairs here?

**Section 5.1**
What was the objective of comparing M-VI and BAL here? I’d appreciate more insight into why M-VI performs so much worse in this setup.

**Section 5.2**
In Figure 9, several agents haven’t converged. Could you provide fully converged results? Additionally, the abstract claims “significant empirical improvement”—could you specify the basis for this assertion?

**Questions:**

See weakness.

---

> ### Author Response · Authors · 2024-11-13
> **Thank you for your comment.**
>
> Thank you for reviewing our paper. We truly appreciate your comments. We understand that your concern is mainly on the clarity of the expositions. The followings are our answers. We will update our manuscript accordingly.
>
> ### Section 4.
> - As the reviewer pointed out, and as stated in the beginnig of Section 4, the motivation to introduce BAL is to theoretically understand the effectiveness of the bounding functions $f,g$. Rather than analyzing the specific choice, e.g. $f(x)=g(x)={\rm tanh}(x)$, we tried to characterized the conditions of $f,g$ that are validated and effective. For this purpose, we found that relating MDAC to Advantage Learning and extending the analyses in the papers [1,2] is helpful. As a result, our analyses not only show the soundness of BAL, but also ensures that Munchausen RL is convergent even when the ad-hoc clipping is employed (by Theorem 2).
> - L224: the definition of (KL-entropy) regularized MDPs is provided at Section 2 (L103 - L109). The initial submission lacks the definition of the soft state value function, $V$, in Section 2; thank you for pointing this out. It is defined as
> $V = \langle \pi, Q\rangle + \tau \mathcal{H}(\pi) - \lambda D_{\rm KL}(\pi\|\mu)$ with $\pi=\mathcal{G}_\mu^{\lambda,\tau}(Q)$, which is defined in L.106. Please see Appendix A of the paper [3] for derivation.
> - L227:
> We have $V(s) = (\mathbb{L}^\alpha\Psi)(s) = \alpha \log \left<1, \exp\frac{\Psi(s,a)}{\alpha}\right> \to \max_{a}Q(s,a)$ as $\alpha\to 0$, because $\Psi_k = Q_k + \beta\alpha\log\mu \to Q_k$
> and $(\mathbb{L}^\alpha\Psi)(s) \to \max_{a}\Psi(s,a)$, the latter of which is a property of log-sum-exp function $\mathbb{L}^\alpha$ and it holds because the similar arguments as Section 5.2 of [4] applies to $\mathbb{L}^\alpha$ as well. We will add a formal proof of this argument in Appendix.
> - L239: Eq.(7) is derived by simply using $\alpha\log\pi_{k+1} = A_k$ for Eq.(6) and excluding the error term $\epsilon_{k}$.
> - L245: As Eq.(7) suggests, the term $\beta f (A_k)$ is added to the soft Bellman operator. Since the advantage $A_k = \Psi_k - V_k$ is  always non-positive from $V_{k}(s) = \left(\mathbb{L}^\alpha\Psi\right)(s) \ge \max_{a}\Psi(s,a)$, which is a property of log-sum-exp function $\mathbb{L}^\alpha$, the re-parameterized action value $\Psi_k$ is decreased by the term $\beta f (A_k)$. Since $\mathbb{L}^\alpha$ is a "soft" max function, the reduction is smallest at the optimal action $\arg\max_{a}\Psi(s,a)$.
> - L247: Here, the "successor" simply means the "next" state action pairs after the state transition. Since the both sides of Eq.(7) is a function of a state-action pair $(s,a)$, and the state transition operator $P$ is applied to $g(A_k)$, the decreased entropy bonus is evaluated at the successor/next state $s' \sim P(\cdot|s,a)$.
>
> ### Section 5.1
> - Thank you for seeking the motivation of the experiment in Section 5.1. Our initial submission is missing this point as well. As discussed in the beginning of P.6 in [3], the larger the value of $\beta$ is, the slower the initial convergence of MDVI gets, and thus M-VI as well. Since the reduction of the misspecification error by BAL is particularly effective when $\Psi_k$ is far from the optimal, we can expect that BAL is effective especially in earlier iterations. This claim is indeed supported by the result provided by Figure 4 in Section 5.1.
>
> ### Section 5.2
> - In the abstract, we claimed that MDAC's "empirical performance is significantly boosted by bounding the values of actor's log-density terms in the critic's loss function". This claim is based on the experimental results provided in Figure 1 and Figure 5, as the reviewer kindly stated that "I found the performance gap between non-bounded (identity) and bounded (tanh) log-policy to be surprisingly substantial", as one of the strengths of our paper.
> - We believe that our current experimental results are enough to show the effectiveness of MDAC over the baselines, in terms of the lengths of the experiments. This is because, Figure 7 indicates that MDAC reaches reasonable performance much faster than the baselines, and Figure 9 indicates that MDAC is effective especially in larger state-action domains. We would like to remark that it is common to report the experimental results in which not all the agents have reached to the convergence (for example, please see Figure 5 of TD3 paper [5] and Figure 1 of SAC paper [7]).
>
> [1] Bellemare+, Increasing the action gap: New operators for reinforcement learning, AAAI, 2016.
> [2] Zhang+, Robust action gap increasing with clipped advantage learning, AAAI, 2022.
> [3] Vieillard+, Leverage the Average: an Analysis of KL Regularization in Reinforcement Learning, NeurIPS, 2020.
> [4] Asadi & Littman, An Alternative Softmax Operator for Reinforcement Learning, ICML, 2017.
> [5] Fujimoto+, Addressing function approximation error in actor- critic methods, ICML, 2018.
> [6] Haarnoja+, Soft actor-critic algorithms and applications, arXiv, 2018.

---

> ### Author Response · Authors · 2024-11-19
> **Update regarding your concerns**
>
> We updated the manuscript. Please kindly check the updated version. The followings are modifications regarding your concerns.
> - __L218-L225__ motivates to introduce BAL and states the purpose of analyses.
> - __L113-L135__ formally defines the soft state value $V$.
> - The descriptions in __Section 4.1__ are largely improved. We thank the reviewer again that $V(s) \to \max_{a}Q(s,a)$ was a typo; $\alpha\log\pi$ does not vanish in the limit $\alpha\to0$. The statement of Theorem 1 and L355-L366 were modified. The claims are still valid, because we only use the fact $(\mathbb{L}^\alpha\Psi)(s) \to \max_{a}\Psi(s,a)$. The properties of $\mathbb{L}^\alpha$ are proven in __Appendix A.1__.
> - __L397-L402__ motivates the gridworld environment in Section 5.1.

---

> > ### Comment · Reviewer_tZxa · 2024-11-21
> > **Thank you for your responses**
> >
> > Thank you for addressing the comments. Based on your responses, I am raising the score to 6.

---

> ### Author Response · Authors · 2024-11-22
> **Thank you for your feedback and raising the score.**
>
> We sincerely thank your feedback and raising the score. We would like to appreciate your first comments again, which largely improved our expositions.

---

### Official Review · Reviewer_DZWM · 2024-11-02

**Soundness:** 2
**Presentation:** 3
**Contribution:** 3
**Rating:** 6
**Confidence:** 4

**Summary:**

This paper proposes mirror descent actor-critic (MDAC), extending the Munchausen RL proposed in the discrete action case to continuous actions. To address the issue of ill-behaved log density when using off-policy data, the paper suggests bounding the added log density term and hopes that the modified term would provide benefits during learning. Empirically, the suggested ad-hoc fix to the ill-behaved log density term is demonstrated to fix the issue and performs better than the baseline without the added term. Theoretically, the paper studies the bounding strategy in the tabular case and shows that the corresponding value iteration algorithm converges while also providing arguments for bounding over not bounding.

**Strengths:**

The strengths of the paper include its originality, clarity, and significance:
1. The originality of the paper is one of its strengths. Although the technique of bounding the log density term in the off-policy case is ad-hoc and not entirely novel (as the original Munchausen RL also has a similar variant), the theoretical results are new to my knowledge.
2. The paper is mostly clear. It has clear writing and is easy to follow. It also covers most of the important related works.
3. The studied problem, continuous control, is of significance in the RL literature. Mirror-descent-based algorithms are also an important and interesting approach that enjoys theoretical motivations. Thus, the paper might be of interest to many RL researchers.

**Weaknesses:**

Despite having many strengths, this paper is weak in the following areas:
1. The empirical evaluation is limited. The main empirical results only include experiments on six MuJoCo environments and show only marginal improvements over the baseline. There are many other commonly used continuous control benchmarks (e.g., DeepMind Control Suite and Omniverse Isaac Gym environments), including some experiments for these environments that would strengthen the paper, especially if there is a larger improvement to be seen.
2. This is a minor point, but the paper is also weak in its literature discussion. While there are many approaches based on the idea of mirror-descent, this paper only discusses relevant work that follows Munchausen RL. If it can include some discussion of alternative mirror-descent-based approaches, especially how they differ from the studied approach, it could improve the contextualization of the paper.

**Questions:**

Here are some questions that might affect the evaluation:
1. Why would there be a significant difference between *clip(x, -1, 1); current* and *clip(x, -1, 1); successor* in Figure 8? Shouldn’t it be quite small, given that consecutive transitions should be included in the replay buffer?
2. Why is the clipping frequency of *clip(x/10, -1, 1)* so low, given that the log density term is shown to be so large in Figure 3?

Other minor comments:
* There isn’t a definition of $A_k$ in Line 235. Is it $A_k=\alpha\log\pi_{k+1}$?
* Typo in Line 262: genral -> general
* It would be better to clarify the tradeoff mentioned in Line 357. The sentences read incomplete to me.

---

> ### Author Response · Authors · 2024-11-13
> **Thank you for your comment.**
>
> Thank you for reviewing our paper. We truly appreciate your comments. Your positive responses to Strengths is very encouraging, and your concerning comments are very constructive. The followings are our responses at this moment. We will update the manuscript accordingly.
>
> ### On Weaknesses
>
> 1. We would like to argue that our experimental results on Mujoco is not marginal. Figure 7, which is the aggregated result provided for reliable benchmarking, indicates that MDAC requires only half amount of samples to reach reasonable performance compared to SAC. However, as you point out, the experiments in larger domain would strengthen our paper. We would like to perform such experiments in the discussion period, as long as the time and the computational resource constraints allow.
>
> 2. We totally agree that our literature discussion is focused on the works relevant to Munchausen RL, and weak in mirror descent based RL methods. We will also conduct additional literature search and update the manuscript to better contextualize our paper.
>
> ### Answers to Questions
>
> 1. The difference of `clip(x, -1, 1); current` and `clip(x, -1, 1); successor` comes from the source of action samples $a$ and $a'$ for which they are evaluated.
> The naive Munchausen bonus term is evaluated for the off-policy $(s,a)$ pairs in the replay buffer. On the other hand, the entropy bonus term is evaluated for the successor $(s',a')$ pairs, where $a'$ is sampled from the `current` actor. Thanks to this "on-policyness" of $a'$, the explosion of the `successor` entropy bonus term is not as severe as the `current` Munchausen bonus term.
>
> 2. Figure 3 shows the quantities for the case __without__ the bounding functions, i.e. $f=g=I$. In this case, there is a negative feedback originated from the TD learning and the actor-critic architecture; (1) the magnitude of log policy terms hinder the critic's loss, (2) the critic's estimate explodes, (3) the inaccurate critic's estimate hinders the policy learning, and (4) the magnitude of the log policy terms get bigger again. On the other hand, if we introduce the bounding functions, the log policy terms much less hinder the critic's estimate. Thus, this negative feedback does not severely happen.
>
> - We also appreciate the minor comments. We will update the manuscript accordingly.
>   - Yes, the definition of $A_k$ is $A_k=\alpha\log\pi_{k+1}$.
>   - L357 would be updated like: $g=I$ always satisfies the sufficient condition of asymptotic convergence (9), but the error of the optimal policy misspecification is less decreased. On the other hand, $g(x)\equiv 0$ largely decreases the error of the optimal policy misspecification particularly when $\Psi_k$ is far from optimal, i.e. in the earlier learning stages, though which possibly violate (9) and hinders the asymptotic performance.

---

> ### Author Response · Authors · 2024-11-19
> **Update regarding your concerns**
>
> We updated the manuscript. Please kindly check the updated version. The followings are modifications regarding your concerns. We believe that your first concerns are addressed, except the empirical evaluation.
> - __L062-L068__ contextualize our paper in the mirror descent literature. If we are missing important related works, it is grateful if the reviewer could point out.
> - __L168-L183__ discusses the explosion of the Munchasen bonus and the entropy bonus term. Figure 2 also addresses your questions on clipping frequencies.
> - __L373-L377__ is an improved description of the trade-off in the choice of $g$.

---

> ### Comment · Reviewer_DZWM · 2024-11-21
> **Thank you for your response**
>
> Thank you for your response and for addressing the minor comments. Here are some follow up:
> ### On Weaknesses
> 1. I agree that the average performance of MDAC at 1.5M steps reaches that of SAC at 3M steps. However, oftentimes, the learning performance of an algorithm gradually plateau after some training. So it would be relatively easy to make such a claim that algorithm A requires half the amount of samples to reach the performance of algorithm B with X samples as long as 1) algorithm A is faster than algorithm B to some degree and 2) X is large enough. Nevertheless, I’m not trying to make an argument to criticize that MDAC is not providing good learning benefits over SAC. What I wanted to comment with Weakness 1 is that, in my opinion, the empirical evidence of MDAC’s effectiveness would be much stronger if either 1) the improvement shown in Figure 7 is based on a large set of environments (but not just six from MuJoCo), or 2) there is some environment(s) where the performance of SAC is terrible or couldn’t even solve it, but MDAC can effectively solve.
> 2. Thank you for adding the additional reference on related work. Modifying the paper according to the following questions is not necessary, but could you shed some light on the difference between different mirror descent frameworks? Do you have any insight on why and when should one consider Munchausen RL over others?
>
> ### On Questions
> 1. Why would the clip ratio be so large in the second half of the training in HalfCheetah? It looks like $\alpha$ is larger in HalfCheetah than in Walker, but their magnitude of $\log \pi_\theta(a|s)$ is similar, and the target entropy is the same $-dim(\mathcal{A})=-6$. Do you have any thoughts about this observation?
> 2. Your explanation makes sense to me. It is possible that if the actor is more stable, $\alpha \log \pi / 10$ could be quite small (it seems to be the case in Figure 2).
>
> In addition, I think Reviewer NWya raises a very interesting point that “things started to destabilize only after $\alpha$ started increasing”. I would like to understand why $\alpha$ might become ill-behaved first, if possible.
>
> Thank you again for your rebuttal and your interesting results.

---

> ### Author Response · Authors · 2024-11-22
> **Thank you for your feedbacks; reply to your questions.**
>
> We thank the reviewer again for the kind and constructive feedbacks. In this reply, let us address the questions in the reviewer's latest reply.
>
> Though we do not have strong evidence to support yet, the followings are our conjecture on the behavior of $\alpha$.
> - The followings are the experimental results acquired so far:
>   - By construction, the direct outcome by bounding is the stabilization of $Q_\psi$ only.
>   - However, as we observe from Figure 2, the bounding stabilizes not only $Q_\psi$ but also $\alpha$.
>   - Thus, the stable behavior of $Q_\psi$ seems the key for stable $\alpha$.
> - On the properties of loss functions:
>   - The $\alpha\log\pi$ is forced to have similar value to $Q_\psi$ through the loss (4).
>   -  On the other hand, though the the loss (5) does not have the explicit notion of the quantities related to reward or value,
> 	from the identity in $\alpha\log\pi=A$ in the tabular case, we can think of the loss (5) as a kind of a constraint to the soft advantage, which requires $\alpha$ to have the value, with which the (scaled) soft advantage matches $\bar{\mathcal{H}}=-{\rm dim}\mathcal{A}$, __regardless__ the scale of reward and values.
> - Give the observations above, we conjecture that, the cause of exploding $\alpha$
> is this target mismatch between (4) and (5); $\log\pi$ tries to stay close to $\bar{\mathcal{H}}$, and $\alpha$ grows large to compensate the scale mismatch with $Q_\psi$. By bounding, since $Q_\psi$ behaves stably, $\alpha$ need not to be aggressive for this compensation.
>
> - Indeed, Figure 14 in Appendix of our updated manuscript shows that, $Q_\psi$ grows faster around 1e5 in HalfCheetah-v4 than Walker2d-v4, and $\alpha$ grows large around this time steps.

---

> > ### Comment · Reviewer_DZWM · 2024-11-25
> > **Follow-up on the behavior of alpha**
> >
> > Thank you for your effort in digging into the observation. What is $A$ in $\alpha \log \pi = A$? I still don’t get why $\alpha$ would try to grow to compensate for the scale mismatch with $Q_\psi$, given that Eq. (5) only contains $\log \pi$.

---

> ### Author Response · Authors · 2024-11-27
> **Additional replies to address your concerns**
>
> Thank you for additional comments and questions. The followings are our answers.
>
> - On limited experimental evaluation:
>   - We agree with your points of experimental evaluation. We are currently conducting additional experiments. We will report the result by the end of the author revision deadline at least.
>
> - On mirror descent perspectives:
>   - __[Wang+ 2019]__ explores off-policy policy gradients in MD view and proposes
> an off-policy variant of PPO.
>   - __[Tomar+ 2022]__ considers a MD structure with the advantage function and the KL divergence, and proposes variants of SAC and PPO.
>   - __[Yang+ 2022]__ incorporates a variance reduction method into MD based RL.
>   - __[Vaswani+ 2022]__ and __[Alfano+ 2023]__ try to generalize the existing MD based approaches to general policy parameterizations.
>   - __[Kuba+ 2022]__ proposes a further generalization that unify even non-regularized RL methods such as DDPG and A3C.
>   - __[Lan 2022]__ proposes a MD method that resembles MDVI, which incorporates both the (Bregman/KL) divergence and an additional convex regularizer, and show that it achieves fast linear rate of convergence.
>   - We think that Munchausen RL is distinct from the above literature in the sense that, it is __implicit__ mirror descent due to the sound reparameterization by __[Vieillard+ 2020b]__. Though this makes it very easy to implement, the control of the policy change is vague, particularly when combined with function approximations. Thus, we think that (1) Munchausen RL based methods are very good starting point to use, and (2) if a precise control of policy change is demanded, another MD methods could be tried.
>
> - Refined conjecture on the behavior of $\alpha$:
>   - Thank you for seeking the clarification. We intended the soft advantage by $A$ in our previous reply. However, we noticed that the phenomenon is much simpler. A naive update of $\alpha$ using the loss (5) and SGD is expressed as
> $\alpha \leftarrow \alpha + \frac{\rho (1-\beta)}{N} \sum_{n=1}^{N}  \left( \log \pi_{\theta}(a_n|s_n) - {\rm dim}(\mathcal{A})\right)$, where $\rho>0$ is a step-size, $N$ is a mini-batch size and
> $a_n\sim\pi_{\theta}(\cdot|s_n)$.
>   - The above expression indicates that, if the average of $\log\pi_{\theta}(a|s)$ over sampled mini-batches are bigger than ${\rm dim}(\mathcal{A})$, $\alpha$ keeps growing. This could be the case when (i) $Q_\psi$ is unstable and (ii) $\log\pi$ is ruined through the loss (4). We will update the manuscript to argue this phenomenon in Section 3.

---

> > ### Comment · Reviewer_DZWM · 2024-12-01
> > **Thank you for your clarification and the additional experiments.**
> >
> > Thank you for your clarification and the additional experiments. Here are some follow up:
> > - On the additional experiments: Both MDAC and SAC perform poorly and are very unstable in the tested new environments. In my opinion, it’s difficult to draw a conclusion from this set of experiments.
> > - On mirror descent perspectives: Thank you for clarifying different perspectives. It would be great to incorporate some of the discussion in the paper.
> > - On the behavior of $\alpha$: The explanation intuitively makes sense, but from the current analysis in Figure 2, we cannot confirm that $\log\pi$ is “ruined” before $\alpha$ starts to diverge - it seems to be in a normal range.
> >
> > Based on the current status of the paper, I maintain my current positive rating and increase my confidence level to 4. If there are clearer results that demonstrate the effectiveness of MDAC, I will further adjust my assessment.

---

> ### Author Response · Authors · 2024-12-03
> **Thank you for your feedbacks.**
>
> Thank you for your feedbacks and increase the confidence of positive rating. The followings are our responses to your latest reply.
> - __On additional experimental results:__
>   - We are keeping running additional experiments on DeepMind Control Suite. By the end of the author reply deadline, we will report the result by comment.
> - __On mirror descent perspectives:__
>   - Yes, we are happy to incorporate the discussion.
> - __On the behavior of $\alpha$:__
>   - We apologize for the incomplete evidence in our latest manuscript. Since Figure 2 only shows the means of the quantities over the multiple sampled mini-batches, $\log\pi$ behaves similar especially for `log_std_min=-2, identity` and `log_std_min=-5, tanh`. We prepared a chart that explicitly compares ${\rm dim}(\mathcal{A})$ and the __distributions__ of $\log\pi(a|s)$ values. The chart clearly supports our latest hypothesis on $\alpha$; we found that
>     1. for  `log_std_min=-5, identity` and  `log_std_min=-2, identity`, $\alpha$ grows bigger especially when $\log\pi(a|s)$ values concentrate to the mean that is bigger than ${\rm dim}(\mathcal{A})$, and
>     2. for  `log_std_min=-5, tanh`, the concentration of $\log\pi(a|s)$ is not observed; the tail of $\log\pi(a|s)$ is heavier especially on the lower side of the distribution, and ${\rm dim}(\mathcal{A})$ is often in the range of one standard deviation.

---

### Official Review · Reviewer_STeN · 2024-11-03

**Soundness:** 2
**Presentation:** 1
**Contribution:** 1
**Rating:** 5
**Confidence:** 3

**Summary:**

This paper proposes to clip the logarithmic term in Mirror Descent Actor Critic (MDAC) when improving the Q value. Experiments on classical benchmarks show that the new approach can attain faster convergence compared to baseline methods. To justify the clip operation, The work argues that it can reduce the upper bound of a certain error term appeared in their convergence analysis.

**Strengths:**

* After applyinig the bound operation, the algorithm is empirically observed to converge faster and achieve higher scores.
* The clip operation is easy to implement.

**Weaknesses:**

* The justification on larger error tolerance for critic value estimation is not valid. Specifically, the paper argues that the proposed algorithm's error term's **upper bound** (the last term in equation (10)) is lower compared to the baseline algorithm (line 345-350). However, lower upper bound does not indicate that the term is lower. Given that the main motivation for the algorithm is its better error tolerance, a rigorous justification is critical.

* The writing needs to be polished; the paper is full of tedious definitions and theorems that are not helpful for readers to understand the main argument. There are many typos as well.

**Questions:**

* My conjecture is that the faster convergence from the clip operation comes from lower variance of the stochastic gradient, which helps stablizing the training?

---

> ### Author Response · Authors · 2024-11-13
> **Thank you for your comment.**
>
> Thank you for reviewing our paper. We sincerely appreciate your comments.
> ### On Weaknesses
>
> - On the justification of larger error tolerance (Theorem 3):
>   - Please kindly check again the requirements to $f,g$, non-positivity of $A_{k}$, non-negativity of $P$, and the definition of the error term ${\Delta^{fg}_{k}}$. The choice $f\ne I$ and $g\ne I$, at least $f\ne I$, indeed decreases the errors and improves the sub-optimality, compared to the case without bounding $f=g= I$. We mentioned the upper bound of the error terms at L345-L350 just to emphasize the possible maximum amount of the error reduction.
>
> - On "tedious definitions and theorems":
>   - Could you kindly clarify which definitions and theorems are not helpful for readers to understand the main argument? Our theoretical arguments are provided to characterizes (1) how we should choose the bounding functions $f,g$ and (2) why they are beneficial, both of which are precisely in the scope of our paper. To be precise, Theorem 1, Theorem 2 and Proposition 1 assert that, BAL is asymptotically convergent if the bounding functions $f,g$ are carefully chosen. Theorem 3 claims that the bounding functions are beneficial in terms of the sub-optimality gap.
>
> ### Answer to Question
> - Since the magnitude of the TD error in critic's loss function is decreased by the bounding functions, we agree that the variance of the stochastic gradient could be also decreased and it contributes to the faster convergence in approximated settings (the experiments in Section 5.2). However, the reviewer's conjecture does not explain the faster convergence of BAL in the model-based tabular setting (Figure 4 for the experiment in Section 5.1), which is supported by our Theorem 3.

---

> > ### Comment · Reviewer_STeN · 2024-11-24
> > **Follow-up Questions**
> >
> > Thanks for your response.
> >
> > After reading the response, I am still at bit confused about why BAL necessarily reduces the error $|\Delta f_k^{f,g}| < |\Delta f_k^{f,I}|$ (line 361), and why the misspecification error is decreased for $f \neq I$ and $g \neq I$, respectively (line 367-372). To me, the proposed BAL seems to adjust the original objective to a new objective that is easier to optimize. However, how good is the new objective is not very clear to me. Could authors provide more details on this part? I am not an expert of the area and may misunderstand some technical details, but I believe characterizing how good the BAL's convergent point is essential for justifying the algorithmic design.
> >
> > I could increase my score if the question can be clearly justified.

---

> ### Author Response · Authors · 2024-11-19
> **Update regarding your concerns**
>
> We updated the manuscript. Please kindly check the updated version. The followings are modifications regarding your concerns.
> - __L360-L361__ explicitly states that the error is reduced.
> - __L218-L225__ motivates to define BAL and states the purpose of analyses.
> - __L212-L213__ is also the statement on the purpose of our analysis.

---

> ### Author Response · Authors · 2024-11-27
> **Thank you for your constructive feedbacks.**
>
> Thank you for your kind and constructive feedbacks, and seeking for clarifications. The followings are our answers.
>
> - On error reduction:
>   - We apologize for the confusion. In our first reply, we stated that "$f\ne I,g\ne I$, at least $f\ne I$, indeed decreases the errors", which was not specific enough. Though $f\ne I$ __always__ decreases the error, we admit that $g\ne I$ does __not always__ reduce the error. The followings are the clarification on the error reduction. We will update the paper accordingly.
>   - First, we remark that __the cross term__ $\left< \pi^*, f(A_{k-1}) \right>$ __has major effect__ on the sub-optimality. Since this term is at most an order of $1/(1-\gamma)$ and the other entropy terms are at most an order of $\log |A|$, the cross term is much larger unless the action space is extremely large. Figure 4 partially supports this difference of magnitude.
>   - __Major error reduction by $f$__:
>     - In __L361__ we state that $|\Delta_k^{f}| \le |\Delta_k^{I}|$ for $\Delta_k^{f} = - \beta \left< \pi^*, f(A_{k-1}) \right> = \left< \pi^*, - \beta f(A_{k-1}) \right>$. From the non-positivity of $A_k$ and the requirement to $f$, we always have $A_k = I(A_k) \le f(A_k)$ point-wise and then $-\beta I(A_k(s,a)) \ge - \beta f(A_k(s,a))$ for all $(s,a)$ and $k$, both sides of which are non-negative. Thus, we have $\left< \pi^*, - \beta f(A_{k-1}) \right> \le \left< \pi^*, - \beta I(A_{k-1}) \right>$ point-wise and therefore we have $|\Delta_k^{f}| \le |\Delta_k^{I}|$. This expression holds for both cases where $\pi^{*}$ is the greedy policy ($\alpha\to0$, __L361__) and the softmax policy ($\alpha>0$, __L369__).
>   - __Additional possible error reduction by $g$__:
>     - Let $\Delta_k^{g} = \left< \pi^*, \beta A^* - \gamma P \left<\pi_k,  A_{k-1} - g(A_{k-1}) \right>\right>$. We have $\Delta_k^{I} = \left< \pi^*, \beta A^* \right>$. Since $\Delta_k^{g}$ consists of the entropy terms, $\Delta_k^{g}$ has much less effect on the sub-optimality. Since $A_{k-1}$ is non-positive, we have $A_{k-1} - g(A_{k-1})\le 0$ from the requirements to $g$. Since the stochastic matrix $P$ is non-negative, we have $P \left<\pi_k, A_{k-1} - g(A_{k-1}) \right> \le 0$, where the l.h.s. represents the decreased negative entropy of the successor state and its absolute value is again bounded by $\log |A|$. Since $A^* \le 0$ also, whose absolute value is an order of  $1/(1-\gamma)$, we __can expect__ that $| \Delta_k^{g} | = | \beta A^*(s,a) - \gamma \left(P \left<\pi_k, A_{k-1} - g(A_{k-1}) \right>\right)(s,a) | \le | \beta A^*(s,a) | = |\Delta_k^{I} |$.
>     - This inequality __does not always__ hold because it depends on the actual magnitude of $A^*(s,a)$ and $\left(P \left<\pi_k, A_{k-1} - g(A_{k-1}) \right>\right)(s,a)$. Though it indeed improves the sub-optimality if this error is decreased for the majority of $k$, even when $g\ne I$ does not decrease the error, its negative effect on the sub-optimality is not substantial.
>
> - On justification of the convergence points:
>   - We agree that the characterization of the biased convergence point is important. In this sense, we admit that our theoretical results are not strong enough; they do not __exactly__ characterize the biased point, but they provide the __ranges__ in which the biased points lie. The reason why we could not specify an __exact__ point is that we do not specify $f,g$, but analyze for a __class__ of $f,g$. The motivation is to "characterize the conditions for bounding functions that are validated and effective", as stated at __L219-224__. However, we remark that the obtained __ranges are reasonable and still useful__ as we discuss below.
>   - Theorem 2 states that, for $g=I$, the lower bound is the __original optimal__ point $V_\tau^*$ and the upper bound is $V_\alpha^*$ with $\alpha=\tau + \lambda$, that is, the __optimal__ regularized state value with a stronger entropy regularization. Importantly, we can choose to directly control these upper and lower bounds by our choice of $\tau$ and $\lambda$, though in practice we chose to optimize $\alpha$ through the loss (5).
>   - Proposition 1 states that, for $g\ne I$ the lower bound gets close to the __un-regularized value__ since $g\ne I$ decreases the entropy bonus, which is also reasonable lower bound and can be controlled by our choice of $\tau$ and $\lambda$.

---

> > ### Comment · Reviewer_STeN · 2024-11-29
> > **Follow-up question**
> >
> > Thank you for the detailed response, which resolves some of my questions on the convergence results.
> >
> > I have an additional question regarding the error reduction of $f$. Based on the your derivation in the last response, it seems that let $f(\cdot) = 0$ directly yields $|\Delta_k^f|=0$. Could you provide more explanation on this observation as well as jusfication on the assumption?

---

> ### Author Response · Authors · 2024-11-29
> **Answer regarding the case $f(\cdot)=0$**
>
> Thank you for seeking the clarification. Your observation is correct; if we set $f(\cdot)=0$, we always have $|\Delta_k^{f}|=0$ and a resemble of the classical error bound for Value Iteration [1] is recovered. This is because the choice $f(\cdot)=0$ is equivalent to omitting the gap-increasing term, which means __no KL regularization__. This fact is indirectly stated at __L253__. As discussed in [2], the larger value of $\beta$ results in MDVI's slower initial convergence, and thus M-VI as well. Though Advantage Learning and M-VI are robust to function approximation errors thanks to the increased action gaps, the gap-increasing nature is not always helpful. If the incorrect advantage $A_k$ pushes down the value of optimal action more than the other sub-optimal actions, this simply hinders the estimation of $\Psi_k$. The term $\Delta_k^{f}$ indeed quantifies this error. MDAC is intended to be robust against both of (1) the function approximation errors and (2) the aforementioned negative aspect of the gap-increasing nature. Our Theorem 3 supports the claim (2) via the analysis of BAL.
>
> [1] Munos, Error bounds for approximate value iteration. AAAI, 2005.
> [2] Vieillard+, Leverage the Average: an Analysis of KL Regularization in Reinforcement Learning, NeurIPS, 2020.

---

> > ### Comment · Reviewer_STeN · 2024-11-29
> > **Thanks for the clarification**
> >
> > Thanks for the response. I appreciate the authors' efforts in addressing my question.
> >
> > The convergence analysis is not very informative; it basically clips the error induced by KL regularization. Theoretically, the algorithm cannot converge to a optimal point as regularized MDVI, given that $f$ and $g$ are tanh, and I believe that the bias will always exist under general setting. The proposed approach can be a good trick to improve the practical performance though.
> >
> > I have increased my score and will keep it at 5.

---

> ### Author Response · Authors · 2024-11-29
> **Thank you for raising the score.**
>
> Thank you for your kind feedback and raising the score. We believe that your original concern on the error reduction argument  is resolved.
> - On convergence results:
>   - Our theoretical results __do not exclude__ the possibility to preserve the optimality. For example, if the ordering of $\Psi_k(s,a)$ in terms of $a$ is preserved, it is possible to attain the optimal policy through the softmax, since the softmax is not injective. Figure 7 indicates that, if $f$ and $g$ are slower saturating functions, the bounding simply reduces to uniform "scaling" in practice. In such a case, the bounding does not change the ordering of $\Psi_k$.
>   - We agree that the bounding possibly incur a bias in general $f$ and $g$. However, recalling the well-known bias-variance trade off in machine learning literature, it is commonly favored to choose a practically helpful method at a cost of reasonable amount of bias. Our theoretical results assure that the possible amount of bias can be controlled by the hyperparameter values.
>
> Anyhow, thank you for acknowledging that the bounding is a “good trick”. This justification is exactly one of the claims of our paper, and we believe that our analyses are informative enough to support this claim; the error is reduced, and the possible amount of bias can be controlled.

---

### Author Response · Authors · 2024-11-19
**We updated the manuscript.**

Dear reviewers, we have updated the manuscript. We reply to each reviewer on corresponding changes. Section 3 was revised most largely with additional experimental results with `log_std_min=-2`. We believe that the revised manuscript better motivates to introduce the bounding.

---

> ### Author Response · Authors · 2024-11-22
> **Paper updated**
>
> Dear reviewers, we have updated the manuscript. We added __Figure__ 14 to Appendix, which is a resemble of Figure 2 for clipped cases, corresponding to Figure 7. Both Figure 2 and also minor-updated; the $\alpha$ follows right y-axis for better readability. The rest of changes are minor issues, such as typo fixing, at this moment.

---

> ### Author Response · Authors · 2024-11-27
> **Manuscript updated**
>
> Dear reviewers, we have updated the manuscript. Please kindly check the paper again.
> - __L179-L190__ in Section 3: the exploding behavior of $\alpha$ is discussed.
> - __L365-L404__ in Section 4: the exposition of the error reduction is improved.

---

> ### Author Response · Authors · 2024-11-28
> **Additional experimental results**
>
> Dear reviewers, we updated the manuscript. Section 5.3 provides additional experimental results in the challenging `dog` domain from DeepMind Control Suite. Figure 15 in Appendix B.2 shows per-environment results. Due to the space limitation, the per-environmental result for Mujoco was moved to Appendix. Though both MDAC and SAC are numerically unstable in `dog` domain, a similar phenomenon to which was reported in [1], MDAC tends to reach better performance than SAC especially in `walk` and `run` tasks. We believe this result is a good show-case of the benefit of KL-entropy regularization in continuous action domains.
>
> [1] Hansen+, TD-MPC2: Scalable, Robust World Models for Continuous Control, ICLR, 2024.

---

### Author Response · Authors · 2024-12-04
**Discussion Summary**

We appreciate again the time and effort of the reviewers in providing constructive criticisms that have helped us to improve the paper.
Since the discussion period comes to a close, let us summarize the major changes we made to the manuscript, and address the reviewers' remaining concerns again.

### On Major Changes

- __Section 1__:
  - We added a literature discussion on mirror descent based RL methods to better contextualize our paper (thanks to Reviewer __DZWM__).

- __Section 3__:
  - We added a experiment with `log_std_min=-2` to better motivate to introduce the bounding to $\alpha\log\pi$ terms. Exploding behavior of $\alpha$ is also discussed as a source of unstable behavior of naive MDAC (thanks to Reviewer __NWya__ and __DZWM__).

- __Section 4.1__:
  - The purpose of theoretical analyses is clarified (thanks to Reviewer __STeN__).
  - The exposition of BAL and its gap-increasing nature are improved (thanks to Reviewer __tZxa__).

- __Section 4.3__:
  - The exposition of the error reduction property is detailed and clarified (thanks to Reviewer __STeN__).

- __Section 5.1__:
  - The purpose to conduct the gridworld experiment is clarified (thanks to Reviewer __tZxa__).

- __Section 5.1__:
  - We added additional experimental results on the challenging `dog` domain from DeepMind Control Suite (thanks to Reviewer __DZWM__).


### On Remaining Concerns
- The cause of exploding behavior of $\alpha$ (Reviewer __NWya__ and __DZWM__):
  - We prepared a chart that explicitly compares ${\rm dim}(\mathcal{A})$ and the __distributions__ of $\log\pi(a|s)$ values. The chart clearly supports our latest hypothesis on $\alpha$: $\log\pi(a|s)$ is ruined by $Q_\psi$ and it gets bigger than the target ${\rm dim}(\mathcal{A})$. We found that
    1. for  `log_std_min=-5, identity` and  `log_std_min=-2, identity`, $\alpha$ grows bigger especially when $\log\pi(a|s)$ values concentrate to the mean that is bigger than ${\rm dim}(\mathcal{A})$, and
    2. for  `log_std_min=-5, tanh`, the concentration of $\log\pi(a|s)$ is not observed; the tail of $\log\pi(a|s)$ is heavier especially on the lower side of the distribution, and ${\rm dim}(\mathcal{A})$ is often in the range of one standard deviation.

- Limitation of experimental evaluation (Reviewer __DZWM__):
  - We conducted additional experiments on the following domains from DeepMind Control Suite. After learning 2M steps, the mean scores of 10 independent runs were:

      |      | `cheetah-run` | `walker-run` | `humanoid-walk` |
      | :----: |   :----:  |   :----:  |   :----:  |
      | SAC  |   862   | __832__ |   214   |
      | MDAC | __895__ |   799   | __460__ |

  - The aggregated result, including four `dog` domains, was consistent to Figure 9 of the latest manuscript. After 2M steps, MDAC's IQM score was higher than SAC, though the lower confidence interval of MDAC has an overwrap with the upper confidence interval of SAC (in the below, CI stands for pointwise 95\% percentile stratified bootstrap confidence intervals).

      |      | lower CI | IQM | upper CI |
      | :----: |  :----:  |   :----:  |   :----:  |
      | SAC  | 0.71 | 0.93 | 1.22 |
      | MDAC | 1.03 | 1.39 | 2.68 |


- BAL's biased convergence for general $f$ and $g$ (Reviewer __STeN__):
  - We believe our asymptotic convergence results are informative enough given the following observations.
    - Our theoretical results __do not exclude the possibility to preserve the optimality__. For example, if the ordering of $\Psi_k(s,a)$ in terms of $a$ is preserved, it is possible to attain the optimal policy through the softmax, since the softmax is not injective. Figure 7 indicates that, if $f$ and $g$ are slower saturating functions, the bounding simply reduces to uniform "scaling" in practice. In such a case, the bounding does not change the ordering of $\Psi_k$.
    - Furthermore, recalling the well-known bias-variance trade off in machine learning literature, it is commonly favored to choose a practically helpful method at a cost of reasonable amount of bias. Our theoretical results assure that the possible amount of bias can be controlled by the hyperparameter values.

---

### Meta-Review · Area_Chair_ovxT · 2024-12-16

**Metareview:**

This paper introduces mirror descent actor critic for reinforcement learning for continuous action spaces. Theoretical analysis is provided. It is also claimed that the "empirical performance is significantly boosted by bounding the values of actor's log-density terms in the critic's loss function". While reviewers support the second half of the statement (bounding the log-density), we are concerned that the first half of the statement is not well backed. In particular, reviewers agree that Figures 14 and 15 in the appendix show only really marginal performance improvement (the aggregated result in Figures 8 and 9 in the main text does not provide full information). Since half of the major claim is not backed, I recommend rejection.

**Additional Comments On Reviewer Discussion:**

The AC-reviewer discussion is fruitful. Three positive reviewers acknowledge the effect of the proposed method in bonding the log-density term but also agree that the empirical improvement is quite marginal. Since the authors motivate the work in the abstract via the lack of performance improvement of prior methods and half of the major claim in the abstract is about empirical performance, I cannot regard this work as a purely theoretical paper and have to take empirical results into consideration, resulting in a recommendation of rejection.

---

### Decision · Program_Chairs · 2025-01-22

Reject